# The TRIM-NHL protein NHL-2 is a co-factor in the nuclear and somatic RNAi pathways in *C. elegans*

Gregory M Davis[1,2,3†], Shikui Tu[4†], Joshua WT Anderson[1,2†], Rhys N Colson[1,2], Menachem J Gunzburg[1,2‡], Michelle A Francisco[5], Debashish Ray[5], Sean P Shrubsole[1,2], Julia A Sobotka[5], Uri Seroussi[5], Robert X Lao[5], Tuhin Maity[5], Monica Z Wu[5], Katherine McJunkin[6], Quaid D Morris[5], Timothy R Hughes[5], Jacqueline A Wilce[1,2], Julie M Claycomb[5]*, Zhiping Weng[4]*, Peter R Boag[1,2]*

[1]Development and Stem Cells Program, Monash Biomedicine Discovery Institute, Monash University, Melbourne, Australia; [2]Department of Biochemistry and Molecular Biology, Monash University, Melbourne, Australia; [3]School of Health and Life Sciences, Federation University, Victoria, Australia; [4]Program in Bioinformatics and Integrative Biology, University of Massachusetts Medical School, Worcester, United States; [5]Department of Molecular Genetics, University of Toronto, Toronto, Canada; [6]Laboratory of Cellular and Developmental Biology, National Institute of Diabetes and Digestive and Kidney Diseases, National Institutes of Health, Bethesda, United States

*For correspondence:
julie.claycomb@utoronto.ca
(JMC);
zhiping.weng@umassmed.edu
(ZW);
peter.boag@monash.edu (PRB)

†These authors contributed equally to this work

Present address: ‡Monash Institute of Pharmaceutical Sciences, Monash University, Victoria, Australia

Competing interests: The authors declare that no competing interests exist.

**Abstract** Proper regulation of germline gene expression is essential for fertility and maintaining species integrity. In the *C. elegans* germline, a diverse repertoire of regulatory pathways promote the expression of endogenous germline genes and limit the expression of deleterious transcripts to maintain genome homeostasis. Here we show that the conserved TRIM-NHL protein, NHL-2, plays an essential role in the *C. elegans* germline, modulating germline chromatin and meiotic chromosome organization. We uncover a role for NHL-2 as a co-factor in both positively (CSR-1) and negatively (HRDE-1) acting germline 22G-small RNA pathways and the somatic nuclear RNAi pathway. Furthermore, we demonstrate that NHL-2 is a bona fide RNA binding protein and, along with RNA-seq data point to a small RNA independent role for NHL-2 in regulating transcripts at the level of RNA stability. Collectively, our data implicate NHL-2 as an essential hub of gene regulatory activity in both the germline and soma.

DOI: https://doi.org/10.7554/eLife.35478.001

## Introduction

The conserved family of TRIM-NHL proteins have emerged as key regulatory points of gene expression to impact a variety of processes including stem cell self-renewal, developmental patterning and cellular differentiation in metazoans (*Tocchini and Ciosk, 2015*). TRIM-NHL proteins contain a N-terminal tripartite motif (TRIM), composed of a zinc finger RING domain, one or two B-box zinc finger motifs (distinct from the RING domain), and a coiled-coil region, in association with six C-terminal NHL motifs (*Figure 1—figure supplement 1*). This combination of domains enables TRIM-NHL proteins to modulate gene expression in versatile ways, ranging from the ubiquitination of protein targets (via the RING domain and associated B-box motifs) (*Ikeda and Inoue, 2012*) to regulating mRNA stability or translation via the NHL domain (*Loedige et al., 2013*). However, the full

repertoire of gene regulatory activities and the developmental specificity of such functions by TRIM-NHL proteins remains to be fully elucidated.

How TRIM-NHL proteins regulate specific mRNAs is only beginning to become clear. The prevailing notion has been that TRIM-NHL proteins interact with mRNA via interactions with other proteins. For instance, the Drosophila TRIM-NHL protein Brat was long thought to recognize maternal mRNAs via an interaction with the RNA binding protein Pumilio (*Sonoda and Wharton, 2001*). However, TRIM-NHL proteins have recently been shown to directly bind to RNA via NHL motifs (*Laver et al., 2015*; *Loedige et al., 2015*; *Loedige et al., 2014*). For some TRIM-NHL proteins, such as Brat, this interaction occurs with a reasonable degree of sequence specificity, with a preference for U rich motifs, and leads to the post-transcriptional and/or translational regulation of target transcripts (*Laver et al., 2015*; *Loedige et al., 2015*; *Loedige et al., 2014*). In addition to binding and regulating RNA directly, several TRIM-NHL proteins have been implicated in miRNA-mediated regulation of transcripts, adding another layer of complexity to their gene regulatory capacity (*Hammell et al., 2009*; *Schwamborn et al., 2009*; *Neumüller et al., 2008*).

NHL-2 is one of five members of the TRIM-NHL family in *C. elegans* (including NHL-1, 2, 3; LIN-41; NCL-1). Thus far, these paralogs have been shown to control pluripotency (LIN-41) (*Tocchini et al., 2014*), oocyte growth and meiotic maturation (LIN-41) (*Tsukamoto et al., 2017*; *Spike et al., 2014*), and inhibit translation (NCL-1) (*Yi et al., 2015*). In comparison, NHL-2 has been shown function the sex determination pathway (*McJunkin and Ambros, 2017*) and also acts as a co-factor in the *C. elegans* miRNA pathway where it is required for proper developmental timing and cell fate progression (*Karp and Ambros, 2012*; *Hammell et al., 2009*). In the miRNA pathway, NHL-2 does not impact the biogenesis of miRNAs but exerts a positive influence on RNA Induced Silencing Complex activity (RISC, including small RNA or miRNA bound to the Argonaute effector, along with accessory proteins) (*Hammell et al., 2009*). NHL-2 associates with the miRNA specific Argonaute effectors, ALG-1/2, and other miRISC co-factors, including AIN-1/GW182 and the DEAD-box RNA helicase CGH-1/DDX6 (*Hammell et al., 2009*). Furthermore, loss of *nhl-2* does not broadly affect miRNA activity, but instead specifically influences miRISC activity associated with two miRNAs: *let-7* and *lsy-6* (*Hammell et al., 2009*). Similarly, the NHL-2 ortholog TRIM32 in *M. musculus* and *H. sapiens* regulates a small number of miRNAs, including *let-7a*, while in *D. melanogaster,* Brat and Mei-P26 have also been shown to act broadly as miRNA co-factors (*Loedige et al., 2014*; *Schwamborn et al., 2009*; *Neumüller et al., 2008*). Individual TRIM-NHL proteins exert positive or negative effects on the miRNA pathway, and thereby control developmental programs (*Loedige et al., 2014*; *Schwamborn et al., 2009*; *Neumüller et al., 2008*). Although the functions of NHL-2 in the soma are linked to miRNA function, what role, if any, NHL-2 plays in the germline remains unclear.

In addition to miRNAs, the small RNA regulatory repertoire in *C. elegans* includes endogenous small interfering RNAs (endo-siRNAs; 22G-RNAs, 26G-RNAs), and Piwi-interacting RNAs (piRNAs/21U-RNAs), which are part of an elaborate surveillance system to regulate homeostasis of the germline transcriptome (reviewed in (*Billi, 2014*; *Youngman and Claycomb, 2014*)). The 22G-RNA family of endo-siRNAs is the most functionally diverse class of germline small RNAs (named 22G because they are generally 22 nucleotides in length, and possess a 5'-triphosphorylated guanosine residue) (*Gu et al., 2009*). It has been proposed that 22G-RNAs are selectively loaded onto either the Argonaute CSR-1 or the WAGO sub-family of AGOs, where they subsequently go on to fulfill distinct gene-regulatory functions (*de Albuquerque et al., 2015*; *Phillips et al., 2015*; *Phillips et al., 2014*; *Phillips et al., 2012*; *Claycomb et al., 2009*; *Gu et al., 2009*). The synthesis of 22G-RNAs relies on a complex containing an RNA dependent RNA polymerase enzyme (RdRP; EGO-1 or RRF-1), in association with the DEAD-box RNA helicase DRH-3 and the dual Tudor domain protein EKL-1 (*Gu et al., 2009*; *Aoki et al., 2007*). Although it remains largely unclear how the 22G-RNAs are specified for their different AGO effectors, one notable feature of the two pathways is that the CSR-1 22G-RNAs are produced solely by the RdRP EGO-1, while the WAGO 22G-RNAs rely on both RRF-1 and EGO-1 for their biogenesis (*Phillips et al., 2014*; *Phillips et al., 2012*; *Claycomb et al., 2009*; *Gu et al., 2009*).

The CSR-1 22G-RNAs are required for normal chromosome organization in the germline and embryos, likely due to genome-wide effects on chromatin and/or transcription (*Wedeles et al., 2013a*). CSR-1 associates with genomic loci of its gene targets, and recent data point to a role for the CSR-1 pathway in promoting the transcription of these germline genes trans-generationally

(*Cecere et al., 2014*; *Conine et al., 2013*; *Seth et al., 2013*; *Wedeles et al., 2013b*; *Claycomb et al., 2009*), as well as post-transcriptionally fine-tuning the expression of target genes in embryos via its endonucleolytic Slicer activity (*Gerson-Gurwitz et al., 2016*). In contrast, the WAGO 22G-RNAs, namely those associated with WAGO-1 and HRDE-1, are required for silencing of endogenous pseudogenes, transposons, and some protein-coding genes both at the transcriptional and post-transcriptional levels. These activities are largely downstream of the piRNA pathway, and function to recognize and silence foreign nucleic acid (*Ashe et al., 2012*; *Buckley et al., 2012*; *Shirayama et al., 2012*; *Gu et al., 2009*). Together, the CSR-1 and WAGO pathways provide an epigenetic memory of self (CSR-1) from non-self (WAGOs) nucleic acid and are critical for maintaining genomic and transcriptional integrity of the germline (*Wedeles et al., 2014*; *Conine et al., 2013*; *Seth et al., 2013*; *Wedeles et al., 2013b*; *Claycomb et al., 2009*).

Here we characterize the role of NHL-2 in the germline and provide evidence that, in addition to its role in the somatic microRNA pathway, NHL-2 is a biogenesis factor in the CSR-1 and WAGO germline 22G-RNA pathways. Like other 22G-RNA biogenesis factors and AGOs, including CSR-1 and WAGO-1, NHL-2 localizes to P granules in the germline. Characterization of *nhl-2(ok818)* mutants reveals phenotypes consistent with loss of CSR-1 pathway activity, including: embryonic lethality, defects in oocyte chromosome organization, and aberrant accumulation of the repressive histone modification, H3K9me2 on germline autosome chromatin. We also found an unexpected role for NHL-2 in the nuclear RNAi pathway, and accompanying temperature-sensitive transgenerational fertility defect. Using an RNAi screen, we demonstrate that *nhl-2* genetically interacts with the 22G-RNA pathway components *drh-3*, *ekl-1*, *cde-1* and *csr-1*, and physically associates with CSR-1, HRDE-1 and DRH-3. High throughput sequencing of small RNAs in *nhl-2(ok818)* mutants reveals a depletion of 22G-RNAs for a subset of CSR-1 and WAGO target genes. Moreover, alterations in the distribution of 22G-RNAs across CSR-1 target genes suggest a mechanism of NHL-2 action in 22G-RNA biogenesis, and implicate NHL-2 in distinct roles for germline versus somatic small RNA pathways. RNA binding assays demonstrate that NHL-2 is a bona fide RNA binding protein that specifically associates with U-rich sequences, and mRNA-seq experiments on *nhl-2(ok818)* mutants support the model that NHL-2 regulates a large number of somatic transcripts via this intrinsic RNA binding activity. Together, our results show that NHL-2 is a key factor in multiple facets of germline and somatic gene regulation.

## Results

### NHL-2 is required for normal germline function and is enriched in germ granules

While *nhl-2* mRNA is reported to be highly expressed in the *C. elegans* germline (*Ortiz et al., 2014*; *Reinke et al., 2004*) its function in this tissue remains relatively unknown. To determine if NHL-2 is required for fertility, we analyzed the brood size of wild-type and *nhl-2(ok818)* deletion null allele (*Hammell et al., 2009*) animals at 20°C, 23°C and 25°C. At all three temperatures, compared to wild-type worms, *nhl-2(ok818)* displayed a significantly reduced brood size, and at 23°C and 25°C there was a significant increase in embryonic lethality (*Figure 1A,B*). Comparable results were obtained when we used a second *nhl-2* null allele (*nhl-2(ma372 ma399)*) (*Figure 1—figure supplement 2A,B*). This temperature-dependent fertility defect has been described for several strong loss of function or null alleles of germline genes, including factors involved in small RNA pathways (*Phillips et al., 2012*; *Gu et al., 2009*; *van Wolfswinkel et al., 2009*; *Wang and Reinke, 2008*) and is likely to reflect temperature-sensitive processes in this tissue.

To determine if there were any gross morphological defects that would lead to infertility, we examined dissected one-day-old hermaphrodite germlines using DAPI staining with DIC and fluorescence microscopy. DIC microscopy revealed no overt morphological differences between *nhl-2 (ok818)* and wild-type germlines (data not shown). However, when we examined the organization of diakinetic oocyte chromosomes in wild-type and *nhl-2(ok818)* animals via DAPI staining, we noted severe defects in *nhl-2(ok818)* mutants at all temperatures. While wild-type diakinetic oocytes displayed six discrete DAPI bodies, indicative of homologous chromosome pairs, *nhl-2(ok818)* worms possessed a range of chromosomal abnormalities including aggregation into disorganized clumps, and greater than six DAPI bodies (*Figure 1C,D*). Similar results were obtained when we examined

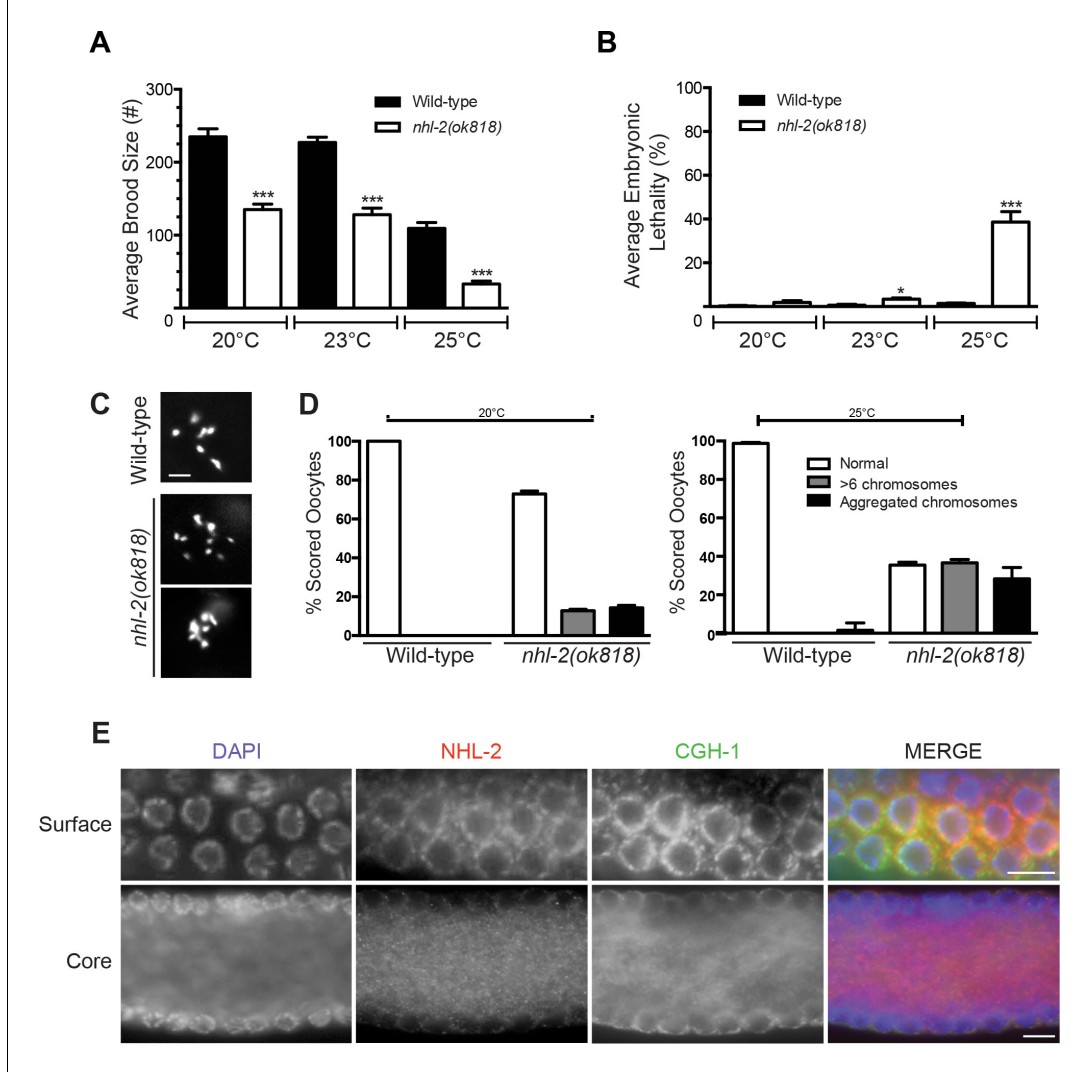

**Figure 1.** NHL-2 is required for fertility and is enriched in germline granules. (**A**) Average brood size and (**B**) embryonic lethality of *nhl-2(ok818)* mutants compared to wild-type at 20°C, 23°C and 25°C. Embryonic lethality is significantly different than wild-type animals at higher temperatures but not significantly different than wild-type at 20°C. Brood size is significantly lower across 20°C, 23°C and 25°C when compared to wild-type animals. \*\*\*=P values <0.001, \*=P values <0.05 error bars represent standard error of mean. (n = 5) (**C**) Representative images of DAPI stained diakinetic oocytes showing abnormal chromosome morphology in *nhl-2(ok818)* when compared to wild-type animals. *nhl-2(ok818)* chromosomes show aggregation or greater than six chromosomal bodies. Scale bar 5 μm, n = 20 (**D**) Quantification of chromosomal defects in diakinetic oocytes in wild-type and *nhl-2 (ok818)* animals at 20°C and 25°C. (**E**) NHL-2 co-localizes with CGH-1 in P-granules (top panels, Surface) and gonadal core CGH-1 granules (bottom panels, Core). DAPI (blue), NHL-2 (red), CGH-1 (green). Scale bar 10 μm. Immunostaining with affinity purified NHL-2 antibody was reduced to background in *nhl-2(ok818)* germlines (*Figure 1—figure supplement 2*).

DOI: https://doi.org/10.7554/eLife.35478.002

The following figure supplements are available for figure 1:

**Figure supplement 1.** Comparison of representative TRIM-NHL proteins.
DOI: https://doi.org/10.7554/eLife.35478.003

**Figure supplement 2.** *nhl-2(ma372 ma399)* mutants display a temperature sensitive reproduction and embryonic lethality defect.
DOI: https://doi.org/10.7554/eLife.35478.004

**Figure supplement 3.** The NHL-2 antibody is specific.
DOI: https://doi.org/10.7554/eLife.35478.005

diakinetic oocyte of *nhl-2(ma372 ma399)* mutants (*Figure 1—figure supplement 2C*). Together these data indicate that NHL-2 is required for normal reproductive capacity and germline chromosome organization.

We next examined where and when NHL-2 is expressed in the germline. To do this, we raised antibodies against the N-terminus of NHL-2 and confirmed that the antibodies were specific for NHL-2 by immunostaining *nhl-2(ok818)* mutant germlines. We then co-stained wild-type germlines for NHL-2 and CGH-1, a germline helicase with which NHL-2 was previously shown to interact (*Hammell et al., 2009*). We found that NHL-2 co-localizes with CGH-1 in the gonad in both perinuclear germ granules, known as P granules in *C. elegans,* and in the cytoplasmic core of the syncytial germline (*Figure 1E- Figure 1—figure supplement 3A*). P granules are cytoplasmic aggregations of mRNA and protein that contribute to germ cell fate, and many RNA regulatory pathways, including those involved in small RNA pathways, localize to these structures (*Voronina et al., 2011*). We also observed that NHL-2 localizes to P granules in early-stage embryos and was quickly lost from the somatic lineages after the 4 cell stage (*Figure 1—figure supplement 3B* and data not shown), consistent with a previous study (*Hyenne et al., 2008*).

Given the interaction of NHL-2 and CGH-1 in the somatic miRNA pathway (*Hammell et al., 2009*), we next examined whether either factor is required for the proper localization of the other factor. First, the absence of NHL-2 does not appear to affect P granule formation, as the P granule markers PGL-1, CAR-1 and CGH-1 localized normally in *nhl-2(ok818)* mutants (*Figure 1—figure supplement 3A* and data not shown). CGH-1 is essential for fertility and localization of several P granule components (*Arnold et al., 2014*; *Sengupta and Boag, 2012*; *Boag et al., 2008*; *Audhya et al., 2005*; *Boag et al., 2005*), however in a *cgh-1* loss of function mutant NHL-2 localized normally, while CAR-1 localized to 'sheet-like' structures in the gonad core (*Figure 1—figure supplement 3C*). These data suggest that although NHL-2 has similar germline localization pattern to CGH-1 and functions with CGH-1 in the soma, NHL-2 may contribute to germline gene regulation through different mechanisms.

## *nhl-2* genetically interacts with the 22G-RNA pathway

To uncover any pathways in which NHL-2 may function to regulate germline development, we conducted a genome-wide RNAi screen for genetic interactions (*Davis et al., 2017*). Out of 11,511 genes screened, we identified 42 genes, that when knocked down in the *nhl-2(ok818)* null background resulted in strong synthetic phenotypes, including sterility, embryonic lethality and larval arrest. Among the candidates are genes encoding factors required for biogenesis of the 22G-RNAs. These include the poly(U) polymerase CDE-1, along with the RdRP complex proteins EKL-1 (a dual Tudor domain protein) and DRH-3 (a Dicer related helicase), which have been shown to physically interact (*Gu et al., 2009*). CDE-1 was specifically implicated in the CSR-1 22G-RNA pathway, while DRH-3 and EKL-1 function in both CSR-1 and WAGO 22G-RNA pathways.

Notably, previous data demonstrated that CSR-1 pathway components display defects in diakinetic chromosome organization similar to *nhl-2(ok818)* mutants (*Claycomb et al., 2009*; *She et al., 2009*; *van Wolfswinkel et al., 2009*; *Nakamura et al., 2007*). To more carefully assess the genetic interaction between *nhl-2(ok818)* and these 22G-RNA factors, we examined diakinetic oocyte defects when we knocked down these genes in wild-type versus *nhl-2(ok818)* animals (*Figure 2A–E*). Because of the link between *cde-1* and *csr-1*, we also examined *csr-1* in these assays. Diakinetic oocyte chromosomal organization was binned into four categories in our assays: 1) normal morphology, where 6 pairs of homologous chromosomes were evident, 2)>6 chromosomes, where univalent (unpaired) chromosomes were present, 3) aggregated chromosomes, where clumping of chromosomes was observed, 4) enhanced aggregation, where chromosomes were tightly clumped. Consistent with previous results, we observed chromosome anomalies when *drh-3*, *csr-1*, *ekl-1* or *cde-1* were knocked down in wild-type animals (*Figure 2A–E*). Importantly, knockdown of each 22G-RNA factor in *nhl-2(ok818)* mutants resulted in enhanced chromosomal aggregation (*Figure 2A–E*) at both 20°C and 25°C. We observed similar enhanced diakinesis defects in *nhl-2(ok818)* double mutants with *csr-1(tm892)* or *drh-3(ne4253)* (data not shown). Furthermore, and consistent with these chromosomal abnormalities, we observed an increase in the embryonic lethality of *nhl-2(ok818)* animals depleted of CSR-1 pathway factors, pointing to a strong genetic interaction (*Figure 2—figure supplement 1*).

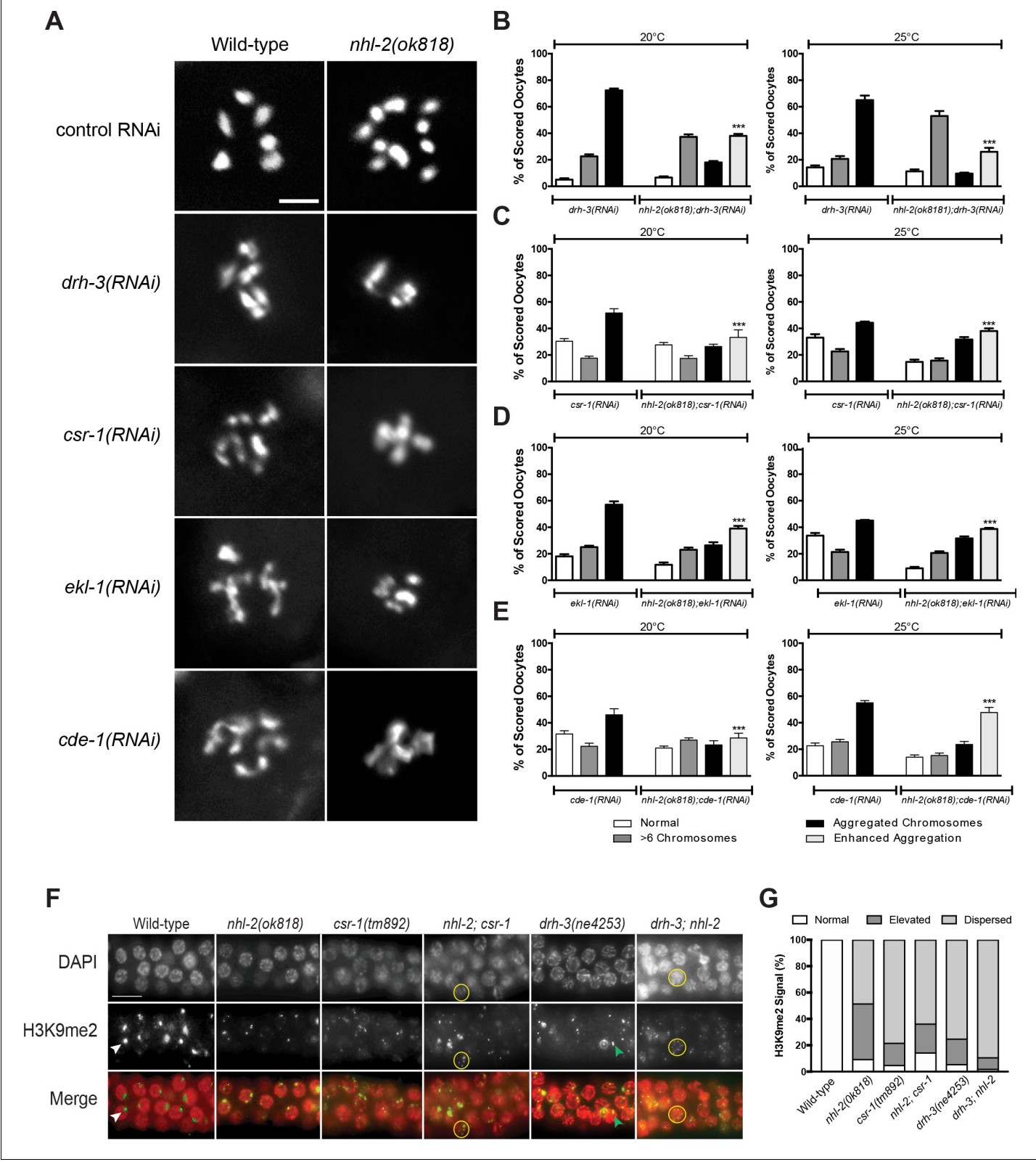

**Figure 2.** NHL-2 genetically interacts with the CSR-1 22G-RNA pathway. (**A**) Representative images of DAPI stained oocyte diakinetic chromosomes showing abnormal chromosome morphology when *drh-3*, *csr-1*, *ekl-1* and *cde-1* were knocked down by RNAi in wild-type and *nhl-2(ok818)* animals. Scale bar 5 μm, n = 90 oocytes. (**B, C, D and E**) Chromosomes from each phenotype were binned in one of four categories: normal, oocytes with >6 chromosomal bodies, aggregated chromosomes, and enhanced aggregation (where aggregation exceeded that observed in wild-type animals, ***=P

*Figure 2 continued on next page*

*Figure 2 continued*

values <0.001). (F) Pachytene region of dissected adult male germlines stained with DAPI (red) and H3K9me2 (green) in wild-type and single and double mutants *nhl-2(ok818)*, *csr-1(tm892)* and *drh-3(ne4253)*. Wild-type males had one strong signal of H3K9me2 (yellow arrowhead), while single and double mutants had increased levels of nuclei with multiple foci (green arrowhead) or dispersed staining patterns (yellow circles) of H2K9me2 staining. Scale bar 50 μm. Over 109 nuclei were score from a minimum of 11 germline. (G) Quantification of H3K9me2 distribution in germ cells from each phenotype scored as normal, elevated, or dispersed from each phenotype. *nhl-2(ok818)* mutants display similar proportions of germ cells with abnormal H3K9me2 markings as the double mutants.

DOI: https://doi.org/10.7554/eLife.35478.006

The following figure supplements are available for figure 2:

**Figure supplement 1.** Synthetic brood size reduction and embryonic lethality between *nhl-2(ok818)* and the CSR-1 pathway.
DOI: https://doi.org/10.7554/eLife.35478.007
**Figure supplement 2.** *nhl-2(ma372 ma399)* mutants show defects in diakinetic chromosome organization and H3K9me2 distribution.
DOI: https://doi.org/10.7554/eLife.35478.008

Another phenotype common among CSR-1 22G-RNA pathway co-factors is the abnormal accumulation of the repressive histone modification, Histone H3, Lysine nine di-methylation (H3K9me2) on autosomes (*She et al., 2009*). Normally, chromosomes that do not possess a pairing partner during meiosis (including the male X chromosome) are enriched for H3K9me2 and transcriptionally silenced in a process termed Meiotic Silencing of Unpaired Chromatin (MSUC). Loss of the CSR-1 pathway leads to aberrant accumulation of H3K9me2 on autosomes and results in homologous pairing defects. Given the synthetic interaction of *nhl-2* with factors in the CSR-1 22G pathway, we examined if *nhl-2* mutants displayed altered H3K9me2 distribution. Using an antibody specific for H3K9me2, we immunostained germlines from wild-type, *nhl-2(ok818)*, *csr-1(tm892)*, and *drh-3(ne4253)* males and compared them to *nhl-2(ok818)*, *drh-3(ne4253)* and *nhl-2(ok818)*, *csr-1(tm892)* males (*Figure 2F*). Consistent with previous reports, *drh-3(ne4253)* or *csr-1(tm892)* mutant males displayed abnormal accumulation of H3K9me2 (*She et al., 2009*), and interestingly, *nhl-2(ok818)* and *nhl-2(ma372 ma399)* males also showed comparable patterns of abnormal H3K9me2 staining (*Figure 2F* and *Figure 2—figure supplement 2*). We went on to quantify the defects in H3K9me2 patterns as follows: (1) normal, where each germ cell displayed one strong H3K9me2 signal (as seen in wild-type), (2) elevated, where germ cells displayed one or more H3K9me2 markings that were greater than that of wild-type germ cells, (3) dispersed, where H3K9me2 markings were distributed on various chromosomes within a germ cell (*Figure 2G*, *Figure 2—figure supplement 2*). This abnormal enrichment of H3K9me2 on autosomes in *nhl-2(ok818)* germ cells suggests that *nhl-2 (ok818)* mutants have errors in MSUC and meiotic synapsis similar to *drh-3(ne4253)* or *csr-1(tm892)* however, double mutants combinations of *nhl-2(ok818)* and *csr-1(tm892)* or *drh-3(ne4253)* exhibited no strong enhancement of this phenotype.

## NHL-2 is required for nuclear RNAi inheritance

Multiple components of small RNA pathways have been shown to be required for nuclear RNAi and germline immortality (*Spracklin et al., 2017*; *Weiser et al., 2017*; *Ashe et al., 2012*; *Buckley et al., 2012*), therefore, we next examined if NHL-2 is required for these pathways via several established assays. First, to examine nuclear RNAi, we focused on the polycistronic pre-mRNA that encodes the non-essential gene *lir-1* and the essential gene *let*-26 (*Bosher et al., 1999*). In wild-type animals, RNAi targeting *lir-1* results in the silencing of the polycistronic pre-mRNA by the nuclear RNAi pathway, resulting in *let*-26 phenotypes of larval arrest and lethality (*Figure 3A*) (*Bosher et al., 1999*). Interestingly, *nhl-2(ok818)* and *nhl-2(ma372 ma399)* animals were resistant to *lir-1* RNAi and closely resembled *nrde-2* and *rde-4* nuclear RNAi mutants (*Figure 3B*, *Figure 3—figure supplement 1A*). This resistance was rescued in *nhl-2(ok818)* worms expressing a GFP-NHL-2 transgene in the soma (*Figure 3B*). Next, because some RNAi factors, such as *hrde-1*, have a temperature-sensitive transgenerational mortal germline (Mrt) phenotypes (*Spracklin et al., 2017*; *Weiser et al., 2017*; *Ashe et al., 2012*; *Buckley et al., 2012*), we tested if *nhl-2(ok818)* also displays this phenotype. Consistent with the Mrt phenotype, after 10–12 generations *nhl-2(ok818)* worms grown at 25°C became sterile (*Figure 3C*) while wild-type worms remained fertile. Similar results were obtained when *nhl-2* was knocked down by continuous administration of RNAi at 25°C and with *nhl-2(ma372 ma399)* (*Figure 3—figure supplement 1B,C*). We next examined if shifting *nhl-2(ok818)* worms grown at 25°C

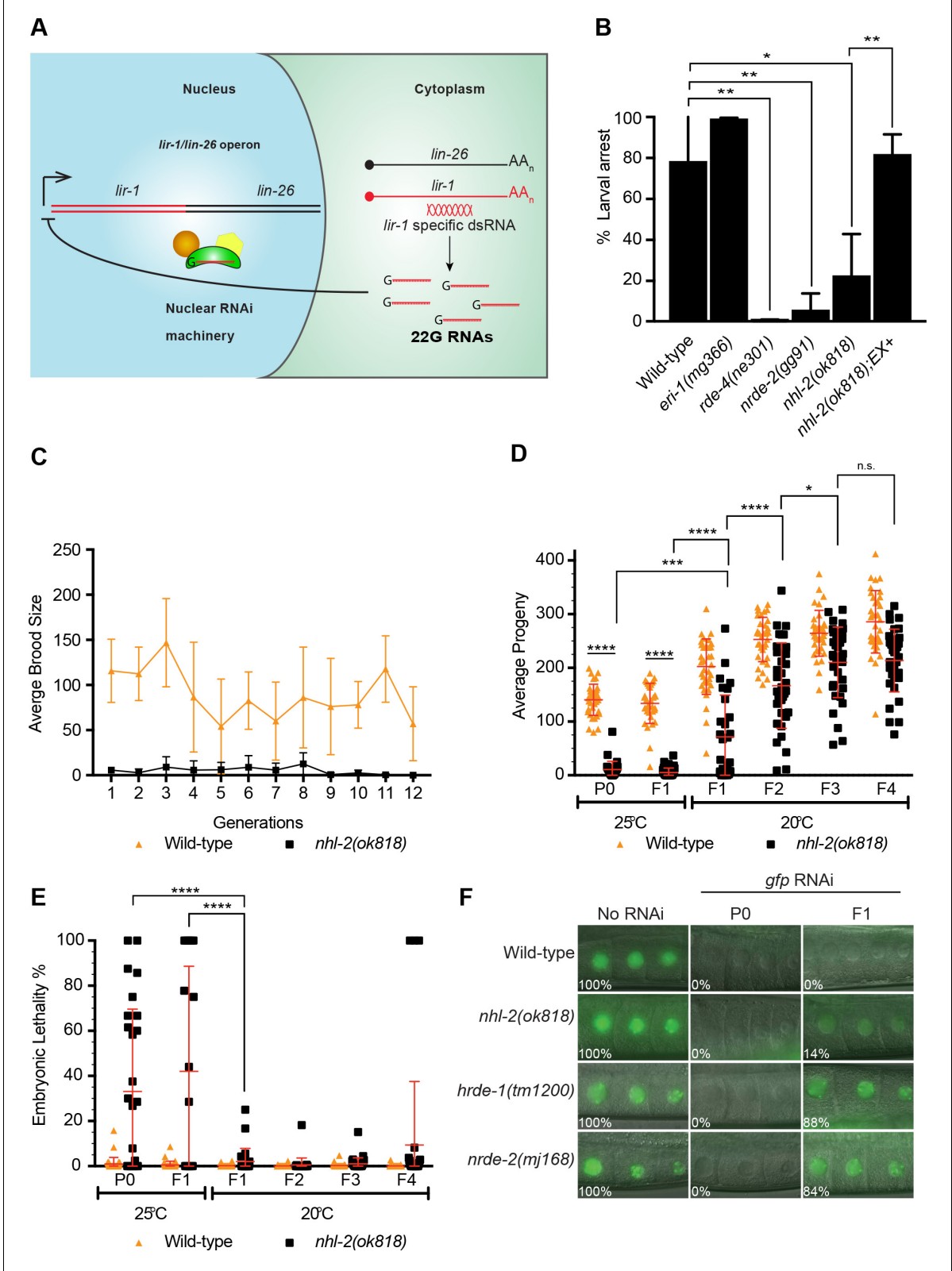

**Figure 3.** Analysis of nuclear RNAi pathways in *nhl-2(ok818)* mutants. (**A**) The nuclear RNAi pathway silences somatic targets. dsRNA targeting the *lir-1* mRNA leads to the generation of 22G small RNAs in the cytoplasm and the nuclear RNAi pathway uses these 22G RNAs to target the endogenous *lir-1/let-26* locus for silencing. (**B**) *nhl-2*(*ok818*) worms are resistant to *lir-1* RNAi. This phenotype was rescued in transgenic *nhl-2(ok818)* worms expressing an GFP-NHL-2 extrachromosomal array (Ex+) under control of the putative *nhl-2* promoter. Percent larval arrest represents mean of two biological

*Figure 3 continued on next page*

Figure 3 continued

replicates ±SD; \*\*=P values <0.001, \*=P values <0.0418, n > 100. (C) *nhl-2(ok818)* worms display a mortal germline phenotype at 25°C. Error bars indicate mean ±SD, n = 6. (D) Analysis of transgenerational brood size and embryonic lethality (E) of wild-type and *nhl-2(ok818)* worms. \*\*\*\*=P values <0.0001, \*\*\*=P values <0.0004, \*=P values <0.0418. Error bars indicate mean ±SD, n = 37. (F) *nhl-2(ok818)* worms have a minor defect in RNAi inheritance at 20°C. Transgenic worms of the indicated genotypes expressing *pie-1::gfp::h2b* were fed *gfp* RNAi for one generation and then grown on OP50 thereafter. Animals were scored for GFP expression using a fluorescence microscope at 63X objective. The percentage of animals expressing GFP is shown, n ≥ 37. Germlines of *nhl-2(ok818)* worms grown at 25°C were unable to be scored due to the disorganised nature of the proximal gonad.
DOI: https://doi.org/10.7554/eLife.35478.009

The following figure supplement is available for figure 3:

**Figure supplement 1.** Analysis of nuclear RNAi pathways in *nhl-2(ma372 ma399)* mutants and *nhl-2(RNAi)*.
DOI: https://doi.org/10.7554/eLife.35478.010

to 20°C could rescue the decline in fertility in subsequent generations. As has been demonstrated for other Mrt mutants (*Spracklin et al., 2017*; *Ni et al., 2016*), when the F1 generation of *nhl-2 (ok818)* worms were moved at the L1 stage to 20°C it took three generations for the brood size to recover to the normal *nhl-2(ok818)* level (*Figure 3D*) and embryonic lethality took a generation to recover to the *nhl-2(ok818)* 20°C level (*Figure 3E*). Similar results were obtained using *nhl-2(ma372 ma399)* mutants (*Figure 3—figure supplement 1D,E*). These findings are consistent the Mrt phenotype observed in mutants in the nuclear RNAi pathway and suggests the decreased brood size observed at 25°C is not simply the result of DNA damage. Finally, we tested whether loss of *nhl-2* impacted the ability of worms to inherit RNAi, as is typical for factors of the nuclear RNAi pathway (*Spracklin et al., 2017*; *Weiser et al., 2017*; *Ashe et al., 2012*; *Buckley et al., 2012*). In this assay, we treated worms expressing a germline GFP marker RNAi food against *gfp* in the P0 generation, and asked whether the progeny remained silenced. We found that NHL-2 was partially required for RNAi inheritance, as 14% of the worms were GFP positive in the F1 generation, compared to 0% for N2 worms and 84% and 88% for *nrde-2(mj168)* and *hrde-1(tm1200)* worms, respectively. Similar results were obtained using *nhl-2(ma372 ma399)* mutants (*Figure 3—figure supplement 1F*). Together these data are consistent with NHL-2 participating in nuclear RNAi and multigenerational epigenetic inheritance pathways.

## NHL-2 physically interacts with CSR-1 and HRDE-1 pathway proteins

Because NHL-2 had previously been shown to physically interact with miRNA Argonautes, we decided to test for any association with the 22G-RNA-associated AGOs CSR-1 and HRDE-1. We first performed co-Immunoprecipitation of endogenous NHL-2 and probed for CSR-1 by western blotting. Indeed, we found that CSR-1 associates with NHL-2 by co-IP in adult hermaphrodites (*Figure 4A*). We then moved on to test for interactions with other members of the CSR-1 pathway by similar experiments, and found an interaction between NHL-2 and the RdRP complex helicase, DRH-3 (*Gu et al., 2009*) (*Figure 4B*). To test for additional protein interactions by a separate method, we incubated purified GST-tagged full-length NHL-2 ((*Figure 4—figure supplement 1A*) with protein lysate and determined whether NHL-2 was able to interact with components of the 22G-RNA biogenesis machinery by western blotting. In these experiments, we observed that GST-NHL-2 associated with both CSR-1 and HRDE-1 from whole adult worm lysates, but not WAGO-1, or other components of the EGO-1, RRF-1 or EKL-1 (*Figure 4C*, data not shown). These data indicate that NHL-2 associates with CSR-1 and HRDE-1 pathway proteins at multiple points of pathway activity (with DRH-3 of the RdRP complex and with AGO effectors of the RISC).

The C-terminal RING domain of TRIM-NHL proteins is often associated with E3 ubiquitin ligase activity and proteasome-mediated protein turnover. To determine whether NHL-2 functions as an E3 ubiquitin ligase that targets proteins of the CSR-1 pathway for degradation, we examined CSR-1 expression in wild-type and *nhl-2(ok818)* one-day-old adult animals. We observed comparable levels of both CSR-1 and DRH-3 protein in wild-type and *nhl-2(ok818)* animals (*Figure 4—figure supplement 1B* and data not shown for DRH-3), suggesting that the association of NHL-2 and CSR-1 is not related to potential E3 ubiquitin ligase activity by NHL-2.

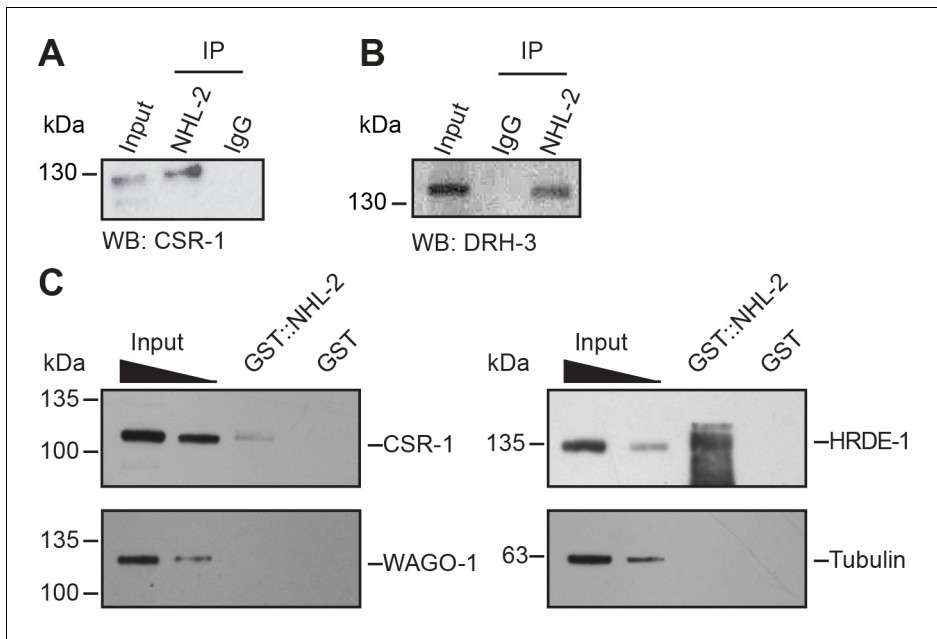

**Figure 4.** NHL-2 interacts physically with the components of the 22G-RNA pathway. (**A**) Western blot using anti-CSR-1 showing no variance between CSR-1 levels in wild-type animals (WT) and *nhl-2(ok818)* mutants when equal amounts of protein are loaded. Tubulin is shown as a loading control (bottom). CSR-1 co-immunoprecipitates with NHL-2. NHL-2 was immunoprecipitated from one-day-old adult hermaphrodite extracts and CSR-1 was detected by Western blot. (**B**) DRH-3 co-immunoprecipitates with NHL-2. NHL-2 was immunoprecipitated from one-day-old adult hermaphrodite extracts and DRH-3 was detected by Western blot. (**C**) GST-pulldown of NHL-2. GST-NHL-2 was able to pulldown CSR-1 and HRDE-1, but not WAGO-1. SDS-PAGE gel of purified GST-NHL-2 is shown in *Figure 4—figure supplement 1*. (**D**) Western blot using anti-CSR-1 showing no variance between CSR-1 levels in wild-type animals (WT) and *nhl-2(ok818)* mutants when equal amounts of protein are loaded. Tubulin is shown as a loading control (bottom).

DOI: https://doi.org/10.7554/eLife.35478.011

The following figure supplement is available for figure 4:

**Figure supplement 1.** The RdRP complex is intact in *nhl-2(ok818)* mutants.
DOI: https://doi.org/10.7554/eLife.35478.012

## NHL-2 is required for maintaining the Steady-State levels of 22G-RNAs

To explore the role of *nhl-2* in the biogenesis or stability of the 22G-RNAs, we conducted small RNA high throughput-sequencing of wild-type and *nhl-2(ok818)* animals at 20°C and 25°C (*Supplementary file 1*, *3*). The overall size and first nucleotide distributions of small RNA species in *nhl-2(ok818)* mutants are consistent with the wild-type controls at both temperatures (*Figure 5—figure supplements 1* and *2*). However, when we began to examine particular classes of small RNAs in each genotype and at the different temperatures, we observed some interesting differences.

First, we observed overall changes in some small RNA populations in both wild-type and *nhl-2 (ok818)* mutants at the higher temperature. There was an overall decrease in 21U-RNAs at 25°C compared with 20°C, for both wild-type and *nhl-2(ok818)* mutants (*Figure 5—figure supplements 1*, *2* and *3*). This may reflect the temperature-dependent nature of the production or stability of these small RNAs. Notably, many mutants in the piRNA and 26G-RNA pathways exhibit temperature sensitive fertility defects that appear to be consistent with an overall decline in these small RNA populations at higher temperatures. 21-23nt siRNA levels (which encompass the 22G-RNA category) increased slightly in both wild-type and *nhl-2(ok818)* mutants at 25°C compared to 20°C, while miRNA levels were relatively unchanged in both strains between temperatures (*Figure 5—figure supplements 1*, *2* and *3*).

Next, we compared wild-type and *nhl-2(ok818)* mutant small RNA populations at each temperature. We observed a consistent decrease in 22G-RNA and 26G-RNA populations in *nhl-2(ok818)*

mutants relative to wild-type at 25°C (*Figure 5—figure supplements 1*, *2* and *3*). Concomitant with the 22G-RNA decreases, we observed an increase in the proportion of miRNAs in *nhl-2(ok818)* mutants. While it is possible that this change in miRNAs is biologically meaningful, it is more likely due to a 'filling in' of the cloning space when small RNA populations are expressed as a proportion of the total reads, as previous reports observed little change in miRNA levels in *nhl-2(ok818)* mutants (*Hammell et al., 2009*). Finally, we observed decreases in the 21U-RNA populations of *nhl-2(ok818)* mutants relative to wild-type at 20°C, which are not evident at 25°C.

Because of their abundance and the link between NHL-2, CSR-1, and the RdRP complex that synthesizes 22G-RNAs (via an interaction between NHL-2 and DRH-3), we examined this class of small RNAs in greater depth (*Figure 5A,B*, *Figure 5—figure supplement 2*). We observed that 573 and 724 genes display a two-fold or greater depletion of 22G-RNAs in *nhl-2(ok818)* mutants relative to wild-type at 20°C and 25°C respectively, with 381 genes in common between the two temperatures (the top-right Venn-pie diagram in *Figure 5A*).

The genes depleted of 22G-RNAs in *nhl-2(ok818)* mutants at 20°C are enriched in WAGO-1 and HRDE-1 target genes (*Figure 5A*), comprising 36.5% (209/573; hypergeometric test q-value or q = 9.6E-80) and 36.6% (210/573; q = 2.1E-83), respectively, with 66 genes shared between the two WAGOs (WAGO-1 targets are as defined in (*Gu et al., 2009*), and HRDE-1 targets are as defined in (*Shirayama et al., 2012*). Consistent with this result, 74.5% (427/573; q = 1.6E-296) of these genes were depleted of 22G-RNAs in a mutant strain that carries mutations in twelve *wago* (12-fold *wago* mutant including mutations in *hrde-1* and *wago-1* [*Gu et al., 2009*]). Notably, 33% (189/573; q = 3.7E-5) of these genes are CSR-1 targets, with only 17 of these genes overlapping with HRDE-1 or WAGO-1 targets (CSR-1 targets are as defined in *Tu et al., 2015*).

Similar to the results at 20°C, 29.3% of the 724 genes depleted of 22G-RNAs in *nhl-2(ok818)* mutants at 25°C are CSR-1 target genes (212/724; q = 3.7E-2), while 33.7% and 37.2% are WAGO-1 and HRDE-1 targets (244/724 and 269/724; q = 2.1E-85 and 4.7E-110; with 139 genes in common). Nearly 78% (563/724, q < 1E-300) of the genes depleted of 22G-RNAs in *nhl-2(ok818)* mutants at 25°C overlap with genes depleted of small RNAs in the 12-fold *wago* mutant. Finally, the majority of genes depleted of 22G-RNAs in *nhl-2(ok818)* mutants at either temperature significantly overlap with genes depleted of 22G-RNAs in RdRP complex mutants, *ekl-1(tm1599)*, *ego-1(om97)*, and *drh-3(ne4253)* (all q-values < 7.3E-195) (*Figure 5A*). The overlap between genes depleted of 22G-RNAs in *nhl-2(ok818)* mutants and the three well characterized germline small RNA pathways, coupled with the overlapping phenotypes of *nhl-2(ok818)* and *ago* mutants point to a role for NHL-2 in germline small RNA pathways overall.

The *glp-4(bn2)* small RNA data set reveals the complete repertoire of germline genes targeted by 22G-RNAs (*Tu et al., 2015*; *Gu et al., 2009*), as these mutants do not possess significant germline tissue, and 5971 genes show a two-fold or greater depletion of small RNAs. Consistent with the role of NHL-2 in fertility and germ cell development (*Figures 1* and *2*), 83.8% (480/573, q = 2.6E-166) and 83.4% (604/724 q = 3.9E-209) of the genes depleted of 22G-RNAs in *nhl-2(ok818)* mutants at 20°C and 25°C, respectively, are also depleted of 22G-RNAs in *glp-4(bn2)* mutants (*Figure 5A*). These data indicate that the genes depleted of 22G-RNAs in *nhl-2(ok818)* mutants are targeted by 22G-RNAs in the germline. Further comparison with the male- and female-specific gonad gene expression data indicated that 58.4% (423/724; q-value = 0.0128) and 61.6% (353/573; q-value = 1E-4) of the genes depleted of 22G-RNAs in *nhl-2(ok818)* mutants at 25°C and 20°C respectively overlap with genes generally expressed in the gonad (meaning that they are expressed in both the spermatogenic or oogenic gonads) (*Figure 5B*). We observed a subtle, but statistically significant enrichment for oogenic genes in the *nhl-2(ok818)* depleted gene sets (16.4%, 119/724 genes at 25°C, q = 2.2E-11, 22.5%, 129/573 genes at 20°C, q-value = 2.7E-24). This observation may simply be reflective of the developmental stage from which the samples were prepared (young adults undergoing oogenesis), but is also consistent with oogenesis defects we observed in *nhl-2(ok818)* mutants.

## NHL-2 is required for 22G-RNA coverage at the 5′ portion of CSR-1 target genes

The decrease in steady-state levels for a subset of 22G-RNAs in *nhl-2(ok818)* mutants could indicate a defect in 22G-RNA synthesis or stability. Given the link between NHL-2 and the RdRP complex, we

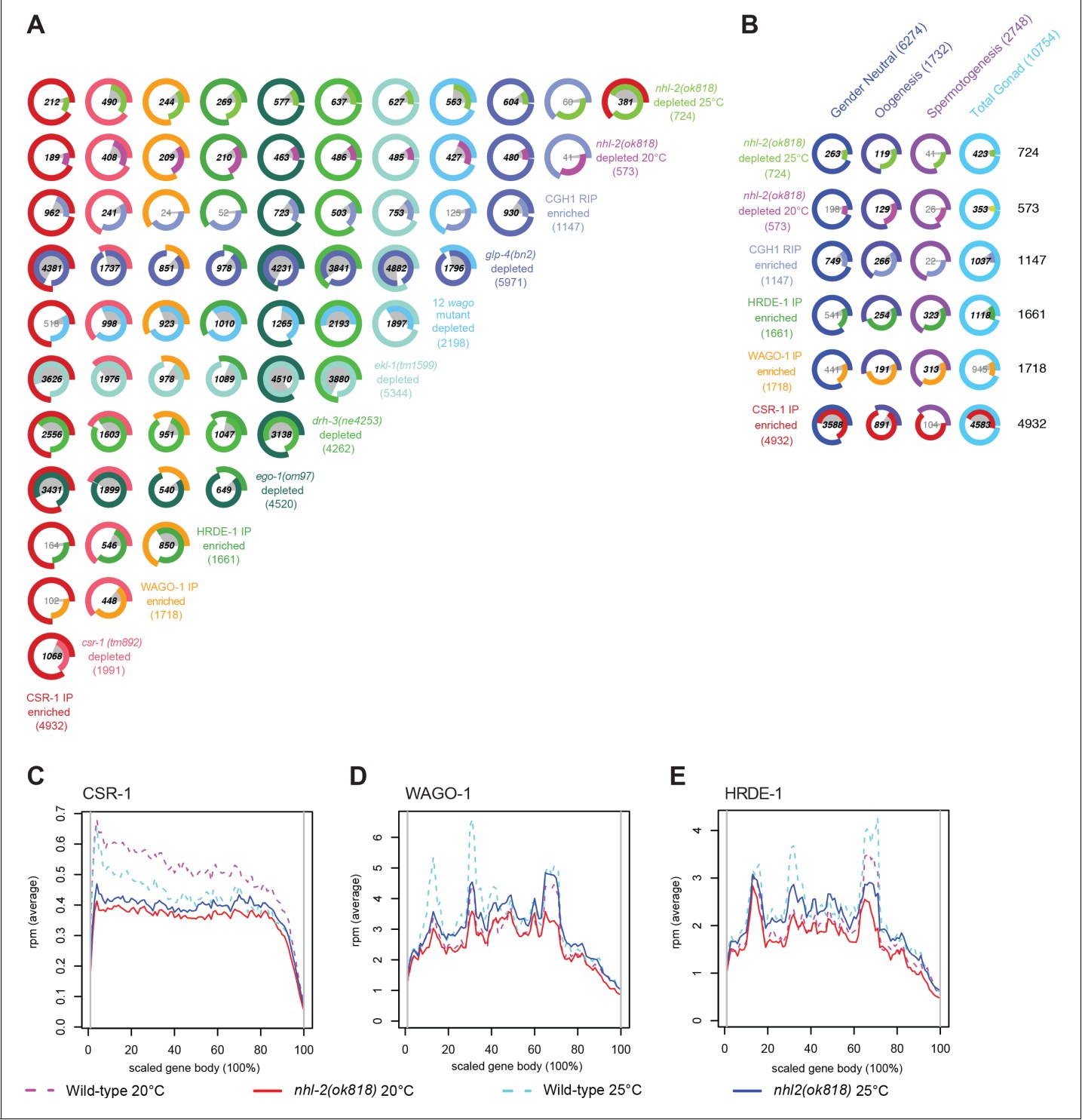

**Figure 5.** Comparison of small RNAs in *nhl-2(ok818)* mutants to other small RNA and mRNA profiles. (**A**) Venn-pie diagrams show comparisons between sets of genes that are the targets of 22G-RNAs. For mutant strains, genes are two-fold or greater depleted of 22G-RNAs in mutants relative to appropriate wild-type controls, with a minimum of ten rpm. For Argonaute IPs, genes are two-fold or greater enriched for 22G-RNAs in the IP relative to a total small RNA input sample, with a minimum of ten rpm. Numbers shown in bold demonstrate statistically significant overlap. (**B**) Venn-pie diagrams show comparisons between 22G-RNA gene targets (as in (**A**), left/rows), and mRNA transcriptome data (bottom/columns) from Total gonads (the union of entire male and female gonad transcriptomes), male gonads (Spermatogenesis), female gonads (Oogenesis), and gender neutral (the overlap between genes expressed in both male and female gonads) genes (*Ortiz et al., 2014*). Numbers shown in bold demonstrate statistically

*Figure 5 continued on next page*

*Figure 5 continued*

significant overlap. (C–E) Distribution of 22G-RNA reads in *nhl-2(ok818)* mutants along the scaled gene bodies are shown for CSR-1 target genes (C), WAGO-1 target genes (D), and HRDE-1 target genes (E) in *nhl-2(ok818)* and wild-type animals.

DOI: https://doi.org/10.7554/eLife.35478.013

The following figure supplements are available for figure 5:

**Figure supplement 1.** Analysis of small RNA populations in *nhl-2(ok818)* mutants and wild-type animals at 20°C.

DOI: https://doi.org/10.7554/eLife.35478.014

**Figure supplement 2.** Analysis of small RNA populations in *nhl-2(ok818)* mutants and wild-type animals at 25°C.

DOI: https://doi.org/10.7554/eLife.35478.015

**Figure supplement 3.** Comparison of abundance (in ppm) for different categories of small RNAs in wild-type (green) and *nhl-2(ok818)* mutants (blue) at 20°C vs. 25°C.

DOI: https://doi.org/10.7554/eLife.35478.016

**Figure supplement 4.** The distribution of 22G-RNAs is reduced at the 5′-end of CSR-1 targets in *nhl-2(ok818)* relative wild-type control samples.

DOI: https://doi.org/10.7554/eLife.35478.017

**Figure supplement 5.** The distribution of 22G-RNAs is reduced slightly at the 3′-end of WAGO-1 targets in *nhl-2(ok818)* relative wild-type control samples.

DOI: https://doi.org/10.7554/eLife.35478.018

**Figure supplement 6.** The distribution of 22G-RNAs is reduced slightly at the 3′-end of HRDE-1 targets in *nhl-2(ok818)* relative wild-type control samples.

DOI: https://doi.org/10.7554/eLife.35478.019

hypothesized that NHL-2 could affect 22G-RNA synthesis to a greater extent than turnover. Thus, we performed a metagene analysis to examine the distribution of 22G-RNAs across the gene body (*Figure 5C–E*, *Figure 5—figure supplements 4*, *5* and *6*). In wild-type worms, 22G-RNAs are distributed across the entire length of WAGO target genes and are present in greater abundance than CSR-1 22G-RNAs. Conversely, CSR-1 target genes generally have fewer 22G-RNAs targeting them overall, with a slight bias in small RNA coverage toward the 5′ end.

For the set of WAGO target genes in *nhl-2(ok818)* mutants relative to wild-type, we noted that the distribution of 22G-RNAs was only mildly changed over the metagene length at either temperature (*Figure 5D,E*, *Figure 5—figure supplements 5* and *6*). We noted, however, a subtle yet statistically significant decrease in the small RNAs toward the 3′ end of WAGO target genes at both 20°C and 25°C, in which the centroid of the 22G-RNA distribution for WAGO-1 targets shifts by 0.5% of the metagene length towards the 5′ end and the distribution for HRDE-1 targets shifts by 1.2% of the metagene length toward the 5′ end at 20°C (Wilcoxon rank-sum test p-value=8.55E-03, p-value=6.09E-03, respectively; *Figure 5*, *Figure 5—figure supplements 5* and *6*).

For CSR-1 target genes, the distribution of 22G-RNAs in *nhl-2(ok818)* mutants was distinct from that in wild-type worms, and striking in comparison to WAGO target genes. At both 20°C and 25°C, there was a significant decrease in the abundance of 22G-RNAs over the 5′ half of the gene in *nhl-2(ok818)* mutants relative to wild-type controls (at 20°C, there was a 33.6% reduction in 22G-RNAs, t-test p-value=1.4E-79; at 25°C, there was a 15.5% reduction in 22G-RNAs, t-test p-value=1.77E-9). Examined a different way, the centroid of the 22G-RNA distribution for the group of CSR-1 target genes shifts by 5.77% of the metagene length towards the 3′ end in *nhl-2(ok818)* mutants relative to the wild-type at 25°C (Wilcoxon rank-sum test p-value=3.67E-29; *Figure 5C*, *Figure 5—figure supplement 4*). These data suggest that NHL-2 may influence the activity and/or processivity of the RdRP complex.

Because our small RNA results indicated a possible role for NHL-2 in RdRP complex processivity or activity, and we had observed genetic and physical interactions between NHL-2 and the RdRP complex via DRH-3, we asked whether NHL-2 was required for the formation or stability of the RdRP complex. Therefore, we tested whether the CSR-1 RdRP complex, as measured by an association between the key components DRH-3, EGO-1, and EKL-1, could properly form in the absence of NHL-2. To answer this question, we immunoprecipitated DRH-3 and probed for EGO-1 and EKL-1, and found that the association between DRH-3, EGO-1, and EKL-1 was maintained in the absence of NHL-2 (*Figure 4—figure supplement 1C,D*). Thus, NHL-2 is not required for the formation or maintenance of the RdRP complex, and may play a different role in the biogenesis of 22G-RNAs. This role could be in the handoff of newly synthesized 22G-RNAs to CSR-1 (because of its physical

association with both CSR-1 and DRH-3, but the lack of a demonstrated interaction between CSR-1 and DRH-3 from the literature (*Duchaine et al., 2006*; *Gu et al., 2009*), aiding the assembled RdRP complex in moving along the RNA template, or in the selection of particular mRNAs as templates for 22G-RNA synthesis.

### The NHL domain of NHL-2 binds RNA

We went on to explore a role for NHL-2 in directly interacting with mRNA transcripts. The NHL domain of the TRIM-NHL protein Brat from *D. melanogaster* was recently shown to be a sequence-specific RNA binding protein, which suggests other NHL domain proteins may also bind directly to RNA (*Loedige et al., 2014*). To determine whether the C-terminal NHL domain of NHL-2 binds RNA, we first generated a structural homology model based on the crystal structure of Brat (*Edwards et al., 2003*). This modeling of the NHL domain revealed the canonical six-bladed 'propeller' characteristic of this domain. It also showed that much of the surface of the protein was positively charged and therefore would likely interact with negatively charged RNA molecules

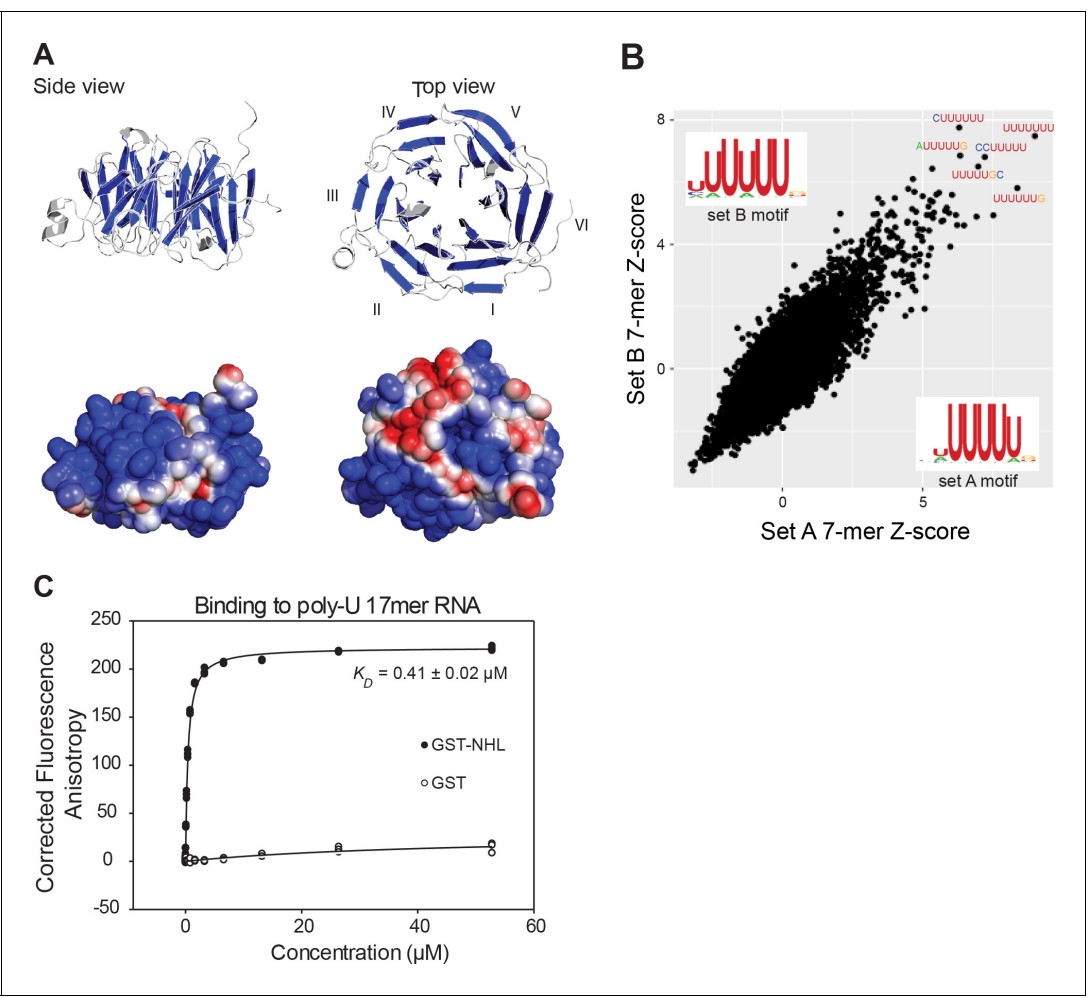

**Figure 6.** The NHL domain of NHL-2 binds RNA. (**A**) Side and top view of the homology-based structure prediction of the NHL-2 NHL domain and its six β-propellers marked blue. Electrostatic surface representation of the NHL-2 NHL domain showing negative regions in blue and positive in red is shown below. The six blades of the β propeller of the NHL domain are numbered (I–VI). (**B**) Identification of RNA-binding motif of purified GST-NHL-2-NHL domain by RNAcompete. The top five high-scoring 7-mers, and the scatter plots, displaying Z scores and motifs for the two halves of the RNA pool (set A and set B) are shown. Sequence logos were derived by aligning the top ten high-scoring 7-mers. (**C**) Quantitative assessment of RNA-binding of GST-NHL-2 NHL domain and GST. The fluorescence anisotropy of reactions contained 1 nM fluorescently-labeled 17mer poly-U single-stranded RNA and increasing concentrations of GST-NHL-2 NHL domain (closed circles) or GST alone (open circles).
DOI: https://doi.org/10.7554/eLife.35478.020

(*Figure 6A*). To explore if the NHL domain of NHL-2 binds to RNA we performed the RNAcompete in vitro binding assay (*Ray et al., 2013*; *Ray et al., 2009*). We expressed and purified a GST-tagged NHL domain of NHL-2 and incubated it with a complex pool of 240,000 30-41mer RNAs. RNAs that co-purified with the GST-tagged NHL domain were identified by microarrays and revealed a strong binding preference for U-enriched RNAs, with a core consensus of UUUU, and preference for U residues 5′ and 3′ to the core (*Figure 6B*). We next examined the binding affinity of the NHL domain for a 5′-Fluorescein labelled 17mer poly-U RNA oligonucleotide using fluorescence anisotropy. In these experiments, GST-tagged NHL domain at a range of concentrations was mixed with poly-U RNA oligonucleotide and allowed to reach equilibrium. These experiments yielded an equilibrium dissociation constant of $K_D = 0.41 \pm 0.02$ µM, and were consistent with a one-site binding model (*Figure 6C*). These data strongly suggest that NHL-2 is a bona fide RNA-binding protein with an ability to bind U rich sequences.

## Steady state mRNA levels are altered independent of small RNA levels in *nhl-2(ok818)* mutants

Because of its link to multiple small RNA pathways and its capacity to bind RNA, we asked whether NHL-2 plays a role in transcript regulation. To test this, we performed mRNA-seq in wild-type (N2) and *nhl-2(ok818)* adult animals at both 20°C and 25°C, with three biological replicates each (*Figure 7A*, *Supplementary file 3*). At 20°C, we observed less extensive changes in steady-state mRNA levels in *nhl-2(ok818)* mutants, with 1014 genes increased and 1630 genes decreased by two-fold or greater. This is in contrast to 25°C, where we identified 3554 genes with two-fold or greater increases in steady-state mRNA levels, and 4370 genes with two-fold or greater decreased steady-state mRNA levels in *nhl-2(ok818)* mutants relative to wild-type (*Figure 7A*). There was a significant overlap of genes up-regulated in *nhl-2(ok818)* between the two temperatures (643 genes; q-value = 6.2E-249). Similarly, there was a significant overlap between down-regulated genes at both temperatures (1326 genes; q-value <1E-300). We then went on to examine the genes with altered expression in more detail.

Based on the previously described role for NHL-2 with *let-7* and *lsy-6* in the miRNA pathway (*Hammell et al., 2009*), we first asked whether predicted targets of these particular miRNAs were de-repressed upon loss of *nhl-2* (*Figure 7B*). Although several lines of data previously suggested that NHL-2 functioned with miRISC in the translational regulation of targets, no genome-wide transcriptome data in *nhl-2(ok818)* mutants were available to test the possibility that NHL-2 and miRISC could impact targets via mRNA stability or turnover. Using TargetScan (Worm Release 6.2, June 2012), we identified 162 predicted targets of *lsy-6*, and 126 predicted targets of the *let-7* family of miRNAs. Only 33 of the 162 predicted *lsy-6* target mRNAs and 29 of the 126 *let-7* family predicted targets were up-regulated in *nhl-2(ok818)* mutants at 25°C. Examining the data the other way around, we asked if the genes up-regulated in *nhl-2(ok818)* mutants were enriched for *let-7* or *lsy-6* predicted targets and found no correlation (data not shown). Overall, these data indicate that miRNA target genes are not regulated by NHL-2 at the level of transcript abundance or stability, and instead, NHL-2 is likely to exert a predominantly translational mode of regulation on these genes.

We next asked whether the genes with altered 22G-RNA levels in *nhl-2(ok818)* mutants were differentially expressed (*Figure 7C*, *Supplementary file 2*). First, we overlapped the sets of genes depleted of 22G-RNAs in *nhl-2(ok818)* mutants at either temperature and were surprised to find only modest effects overall. Of the 573 genes depleted of 22G-RNAs at 20°C, 78 displayed increased steady-state mRNA levels (13.6%; 78/573 genes; q-value = 3.3E-14) and 11 had decreased steady-state mRNA levels (1.9%; 11/573: not significant). At 25°C, 724 genes showed depleted 22G-RNA levels, and of these 210 were up-regulated (29%; 210/724, q-value = 1.8E-13), while 81 were down-regulated (11.2%; 81/724: not significant). Thus, overall, genes displaying altered 22G-RNA levels were not extensively affected at the mRNA level by the loss of *nhl-2*, and the fraction that were affected at the mRNA level tended to be repressed by NHL-2 under wild-type conditions. Overall, these data are consistent with a role for NHL-2 in the biogenesis, but not necessarily the effector steps, of a subset of 22G-RNAs. These data could also point to a role for NHL-2 in regulating the translation of this subset of germline small RNA target genes for which the 22G-RNA levels are altered in *nhl-2(ok818)* mutants, perhaps in a manner similar to the miRNA pathway.

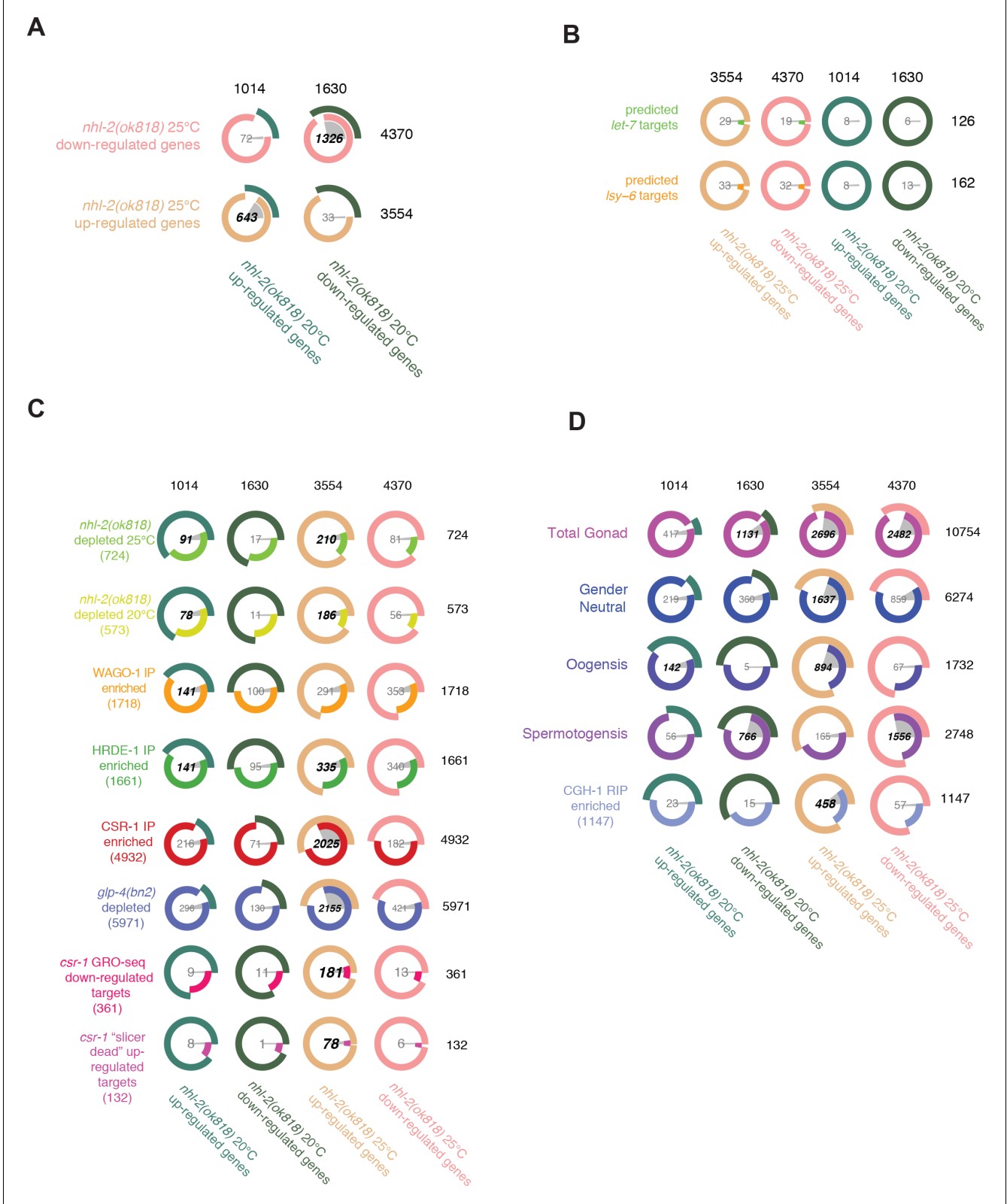

**Figure 7.** Analysis of the 22G-RNAs targeting the *nhl-2(ok818)* mRNA transcriptome. (**A**) Venn-pie diagrams indicate the number of genes enriched in or depleted of 22G-RNAs in *nhl-2(ok818)* mutants relative to wild-type worms at 20℃ and 25℃, as determined using EdgeR. Numbers in bold demonstrate statistically significant overlap. Each row corresponds to a gene set, with its label on the right. Each Venn-pie diagram indicates the overlap between the gene set of its row and the gene set whose label is in its column. (**B**) Venn-pie diagrams show comparisons between the predicted

*Figure 7 continued on next page*

Figure 7 continued

targets of the miRNAs *let-7* and *lsy-6* (determined using TargetScan6), and genes that are mis-regulated in *nhl-2(ok818)* mutants. (**C**) Venn-pie diagrams show comparisons between the genes depleted of 22G-RNAs in *nhl-2(ok818)* mutants and the genes enriched in 22G-RNAs in CSR-1, WAGO-1, or HRDE-1 IP samples with *nhl-2(ok818)* mRNA-seq data. (**D**) Venn-pie diagrams show comparisons between *nhl-2(ok818)* mRNA-seq data and genes depleted of 22G-RNAs in *glp-4(bn2)* mutants (which have very few germ cells) or germline expressed gene mRNA-seq data (*Ortiz et al., 2014*) (as in *Figure 4B*).

DOI: https://doi.org/10.7554/eLife.35478.021

Next, we examined the steady state levels by evaluating the levels of CSR-1 or WAGO pathway target genes in *nhl-2(ok818)* mutants, independent of any changes in 22G-RNA levels (*Claycomb et al., 2009*; *Gu et al., 2009*). In accordance with the opposing roles of the WAGO and CSR-1 pathways in germline gene regulation, we anticipated that the WAGO targets with decreased 22G-RNAs would display increased mRNA levels, while the CSR-1 targets with decreased 22G-RNAs would display decreased mRNA levels in *nhl-2(ok818)* mutants. We observed that a statistically significant subset of WAGO-1 target genes (18%; 141/1718 genes; q-value = 9.9E-8) were de-repressed at 20°C, and 291 out of 1718 WAGO-1 target genes were de-repressed at 25°C (16.9%; q = 0.6790). HRDE-1 target genes also tended to be up-regulated in *nhl-2(ok818)* mutants at both temperatures (141/1661; 8.5% q-value = 9.9E-9 at 20°C, 335/1661; 20.2%, q-value = 9.9E-2 at 25°C). These results are consistent with a cooperative role for NHL-2 in the repression of WAGO target genes, albeit in a small RNA-independent manner.

Unexpectedly, CSR-1 target mRNAs were significantly up-regulated *nhl-2(ok818)* mutants, and this effect occurred specifically at 25°C (2,025/4932; 41%, q-value <1E-300). This was further supported by the significant overlap between the global run-on sequencing (GRO-seq) analysis of *csr-1* partially rescued (hypomorphic) mutant, which identified 361 genes that were down-regulated upon loss of *csr-1* (50%; 181/361 genes; q-value = 4.39E-47) (*Cecere et al., 2014*) and the genes that were up-regulated in *csr-1* 'slicer dead' mutants (59%; 78/132 genes; q-value = 4.16E-27) (*Gerson-Gurwitz et al., 2016*). These data suggest that NHL-2 acts consistently to down-regulate CSR-1 target genes, regardless of whether CSR-1 positively (GRO-seq data) or negatively ('slicer dead' data) impacts these subsets of its targets (notably, only 15 genes overlap between these two groups).

Similarly, when examined the behavior of total set of genes targeted by germline small RNAs using the *glp-4(bn2)* mutant (*Gu et al., 2009*), we found a strong overlap between genes depleted of small RNAs in the *glp-4(bn2)* mutant and genes up-regulated in *nhl-2(ok818)* mutants specifically at 25°C (60%; 2155/3554, q-value <1E-300) (*Figure 7C*). These results are consistent with a small RNA-independent role for NHL-2 in the repression of CSR-1 germline 22G-RNA targets overall. Moreover, the functions of NHL-2 in regulating germline small RNA pathway target genes, especially those of the CSR-1 pathway, appear to be particularly important at high temperature (25°C) or perhaps other stressful conditions. Overall, the changes in steady-state mRNA levels for germline 22G-RNA pathway target genes point to a role for NHL-2 in regulating transcripts via its intrinsic RNA binding capacity.

## Loss of *nhl-2(ok818)* impacts genes involved in Gametogenesis, Signaling, and Chromosome Organization

In an attempt to link alterations in gene expression to changes in phenotype for *nhl-2(ok818)* mutants, we next compared the *nhl-2(ok818)* transcriptome data to spermatogenic versus oogenic gonad transcriptomes (*Ortiz et al., 2014*) (*Figure 7D*). First, we found that a large fraction of genes that were mis-expressed in *nhl-2(ok818)* mutants at both temperatures were represented in total gonad transcriptomes (20°C up-regulated genes 41.1%, 417/1014, q-value = 1; 20°C down-regulated genes 69.4%, 1131/1630, q-value = 7.2E-46; 25°C up-regulated genes 75.9%, 2696/3554, q-value = 5.0E-216; 25°C down-regulated genes 56.8%, 2482/4370, q-value = 3.1E-9), indicating that most of the mis-regulated transcripts are expressed in the germline. Next, we found that genes up-regulated in *nhl-2(ok818)* mutants at 25°C were significantly enriched for gender neutral and oogenesis-associated transcripts (46.1%, 1637/3554; 25.2%, 894/3554; hypergeometric test q-values = 3.8E-100, and 2.8E-259, respectively), while, for those genes up-regulated at 20°C, only oogenesis-associated transcripts were enriched (14%, 142/1014; hypergeometric test

q-value = 9.2E-8). Interestingly, down-regulated genes in *nhl-2(ok818)* mutants at both 20°C and 25°C were enriched for spermatogenesis-associated transcripts (47%, 766/1630; 35.6%, 1556/4370; hypergeometric test q-value = 1.8E-265 and <1E-300, respectively). These data point to an overall misregulation of the germline transcriptome that leads to defects gametogenesis and decreased fertility.

Given that NHL-2 and CGH-1 have been shown to physically interact, we next compared the CGH-1 RNA-IP/microarray data (*Boag et al., 2008*) with the *nhl-2(ok818)* transcriptome data (*Figure 7D*). In the germline, CGH-1 is required to protect specific maternal mRNAs (which overlap with those enriched in the oogenesis and gender neutral transcriptomes) from degradation, and is also involved in translational regulation of some transcripts (*Boag et al., 2008*). We observed a small, but significant, enrichment between CGH-1-associated and up-regulated transcripts in *nhl-2 (ok818)* mutants at 25°C, (12.9%, 458/3554; hypergeometric test q-values = 4.2E-76). These data suggest an antagonistic role between NHL-2 and CGH-1 in the regulation of the stability of these transcripts. To determine if there was any interaction with NHL-2 and other described germline RNP complexes we examined mRNA targets of the cytoplasmic poly(A) polyermase GLD-2 targets and the RNA-binding proteins OMA-1 and LIN-41, all three critical for post-transcriptional regulation of the oocyte-to-embryo transitions (*Tsukamoto et al., 2017*). Similar to CGH-1-associated transcripts, GLD-2 target mRNAs (16.5%, 588/3554, q-value = 1.0E-207) and OMA-1 (20.4%, 725/3554, q-value = 7E-67) and LIN-41-enriched transcripts (13%, 462;3554, q-value = 8E-80) had a small but significant enrichment compared to up-regulated transcripts in *nhl-2(ok818)* mutants at 25°C. This is consistent with the strong overlap between genes expressed during oogenesis and those upregulated in *nhl-2(ok818)* mutants at 25°C. These data suggest that NHL-2 does not regulate the majority of mRNAs found in the key pathways governing the oocyte-to-embryo transitions, however, we cannot rule out any translational effects of NHL-2 in this context.

When we performed Gene Ontology (GO) analysis, we found that the genes that are down-regulated in *nhl-2(ok818)* mutants at 20°C and 25°C shared consistent sets of GO terms, and are strongly enriched in cuticle/collagen proteins, kinases and phosphatases, and spermatogenesis proteins (FDR = 6.1E-20 *protein kinase, core*, 7.9E-25 *phosphatase activity*, and 1.5E-31 *major sperm protein*, respectively). Genes that are up-regulated in *nhl-2(ok818)* mutants at 20°C are weakly enriched in signaling molecules and oxidative metabolism (FDR = 1.4E-12 *signal peptide*, and 2.8E-3 *oxidation reduction*, respectively), and differ from the GO terms observed for the genes up-regulated at 25°C, which were enriched for cell cycle, *kinetochore*, *RNA-binding* and *DNA replication* and *damage repair* (FDR = 5.5E-35 cell cycle, 3.7E-12 *kinetochore*, 1.1E-16 *RNA binding*, and 3.7E14 *DNA replication* and 9.6E10 damage repair GO terms. In addition to analyzing the complete sets of up-regulated or down-regulated genes in the *nhl-2(ok818)* mutants, we also performed GO analysis on sets of transcripts that were expressed in the gonad, and observed comparable results (data not shown). Collectively, our data point to a role for NHL-2 in regulating the stability of a large fraction of gonad/germline transcripts that are involved in spermatogenesis, cellular signaling cascades, cuticle formation, and kinase/phosphatase activities, and chromosome organization. NHL-2 impacts these mRNAs both positively and negatively, and likely utilizes the intrinsic RNA binding properties of its NHL domain, perhaps in association with other protein binding partners, to do so.

## Discussion

TRIM-NHL proteins have been shown to play a variety of crucial roles in the context of the proliferation versus differentiation decision in metazoans. With modular and varied domains, TRIM-NHL proteins can function as E3 ligases as well as sequence-specific RNA binding proteins (*Schwamborn et al., 2009*; *Kudryashova et al., 2005*). TRIM-NHL proteins also engage the miRNA pathway, whereby, remarkably, their functions relate to only a few miRNAs, and impact the efficacy of the RISC both positively and negatively. With their intrinsic RNA binding activity, TRIM-NHL proteins could regulate RNA directly or function at various steps and in virtually any small RNA pathway.

NHL-2 is one of five TRIM-NHL proteins in *C. elegans* (paralogs include NHL-1,–3, LIN-41, and NCL-1). NHL-2 has been shown to modulate miRISC via two specific miRNAs, *let-7* and *lsy-6*, which act in the soma to regulate developmental timing and cell fate transitions in multiple tissues. Recently it was also shown that NHL-2 is also required for sex determination, although the mechanism is unclear (*McJunkin and Ambros, 2017*). In spite of these intriguing roles for NHL-2 in the

embryo and soma, little is known about its functions in germline development. Here, we set out to explore a role for NHL-2 in the germline and in germ cell development. We found that NHL-2 is required for proper germline chromatin organization and wild-type levels of fertility at high temperatures and for the somatic nuclear RNAi pathway. We also identified the AGOs CSR-1 and HRDE-1 and the RdRP component DRH-3 as genetic and physical interactors of NHL-2. High throughput sequencing of small RNA populations in *nhl-2(ok818)* mutants revealed an additional role for NHL-2 in the WAGO-1, and HRDE-1 22G-RNA pathways, but as previous data suggested, little biologically meaningful perturbation in the overall miRNA population. Binding assays confirm that NHL-2 is a bona fide RNA binding protein, and examination of the mRNA transcriptome by mRNA-seq points to NHL-2 as a post-transcriptional regulator of a substantial set of mRNAs involved in signaling, phosphorylation and transcription independent of its small RNA activities. Together, our data, implicate NHL-2 as a regulator of mRNA stability for a significant portion of the genome, a likely translational regulator of miRNA targets, and a biogenesis factor and/or possible translational regulator of targets in the CSR-1 and WAGO 22G-RNA pathways.

We first identified a link between NHL-2 and the CSR-1 pathway by genetic and phenotypic studies, in which loss of CSR-1 pathway factors enhanced the aggregation of diakinetic oocyte chromosomes in *nhl-2(ok818)* mutants. Loss of *nhl-2* also led to increased levels of H3K9me2 in pachytene germline nuclei and a spreading of this heterochromatin modification onto autosomes, where it is not normally observed. This phenotype is consistent with loss of CSR-1 pathway members, providing another phenotypic link between NHL-2 and the CSR-1 pathway. At this time, we do not entirely understand why this phenotype emerges in *nhl-2* or *csr-1* pathway mutants. It is possible that CSR-1 is not properly recruited to its target genes, due to mis-regulation of CSR-1 target transcripts in *nhl-2(ok818)* mutants. This, in turn, could disrupt the formation or maintenance of euchromatin at these loci and allow for the mis-direction of chromatin modifiers throughout the genome, as observed in *csr-1* mutants (Christopher Wedeles and Julie Claycomb, unpublished results). This leads to the aberrant accumulation of histone modifications throughout the genome, which could impact chromosome structure. Future ChIP-seq studies for histone modifications and CSR-1 recruitment in *nhl-2 (ok818)* mutants will enable us to address this possibility. Alternatively, and based on the GO analysis, this chromosome organization defect could result indirectly from alterations in the levels of key transcripts associated with chromosome organization and metabolism, as has been proposed for CSR-1 (*Gerson-Gurwitz et al., 2016*).

It was somewhat surprising to observe little correlation between the genes depleted of 22G-RNAs and genes with altered mRNA levels in *nhl-2(ok818)* mutants. Based on the known regulatory functions of these pathways we expected there would be a slight decrease in the level of CSR-1 target genes for which the 22G-RNAs were depleted, and an increase in the set of genes targeted by WAGO-1 or HRDE-1 for which the 22G-RNAs were depleted. Instead, we observed little change in the steady-state levels of transcripts with depleted 22G-RNAs, indicating that NHL-2 is involved in the translation of these genes, or that there is a different role for NHL-2 with regard to these genes in the 22G-RNA pathways.

The lack of correlation between mRNAs with altered levels and changes in 22G-RNAs raises another possibility: that NHL-2 is mainly involved mainly in the biogenesis of a subset of the 22G-RNAs. This model seems plausible for several reasons. First, DRH-3 and NHL-2 physically interact by co-IP. Second, our metagene analysis of the distribution of 22G-RNAs along the length of target mRNA transcripts is similar to the pattern observed for *drh-3(ne4253)* mutants, in which the 22G-RNAs are reduced along the length of the gene body, with most significant decreases present at the 5′ end of the transcript. This pattern is consistent with a role for NHL-2 in the processivity or activity of the RdRP complex on a subset of CSR-1 targets (*Figure 8*). The RNA binding activity of NHL-2 points to a model whereby NHL-2 could help to identify particular mRNAs as candidates for 22G-RNA synthesis. Furthermore, because NHL-2 also associates with CSR-1 and HRDE-1 as well as DRH-3, it could also act as a chaperone required efficient handoff of 22G-RNAs from the RdRP complex to the Argonaute (*Figure 8*).

This potential role for NHL-2 with the RdRP complex is noteworthy for several reasons. First, because TRIM/NHL proteins have thus far only been implicated in the effector step of miRNA pathways, this is the first indication that NHL-2 (and thus TRIM/NHL proteins) could also be involved in the biogenesis of endo-siRNAs. Second, we still have relatively little insight into the factors that route particular transcripts into the 22G-RNA pathways, and NHL-2 provides an attractive candidate

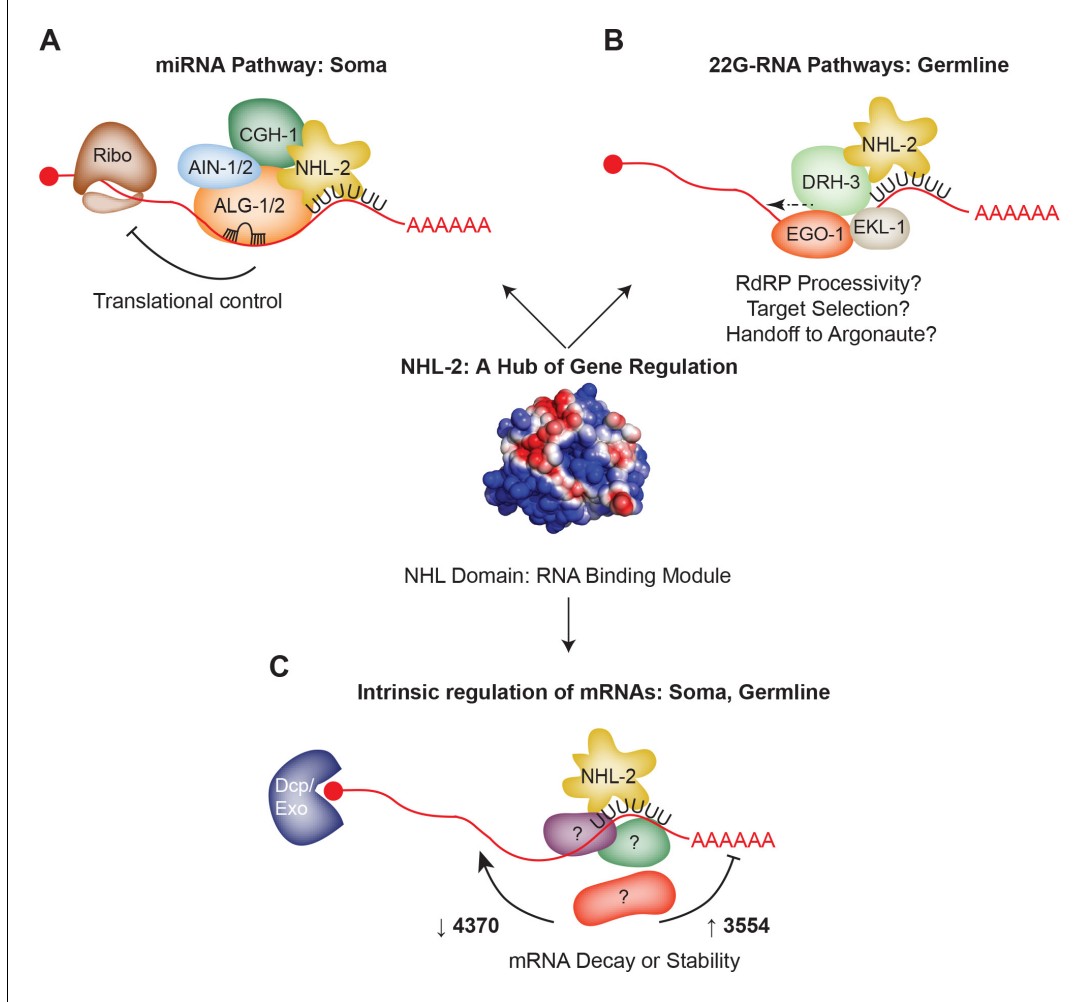

**Figure 8.** Model: NHL-2 acts as a hub of gene regulatory activity. (**A**) NHL-2 interacts with the miRNA pathway in the soma, where it may regulate the translation of *let-7* and *lsy-6* targets. The NHL RNA binding domain of NHL-2 may help to reinforce target selection of miRISC in this pathway. (**B**) In the germline, NHL-2 interacts with the 22G-RNA pathway, specifically via genetic and physical interactions with the RdRP helicase DRH-3. In this capacity, NHL-2 could impact the processivity or reloading of the RdRP complex on mRNA templates, and/or could aid in target mRNA selection via its RNA binding capacity. Owing to its genetic and physical interaction with CSR-1, NHL-2 could also be involved in the handoff of 22G-RNAs from the RdRP complex to Argonaute. If NHL-2 mediates gene regulation via 22G-RNAs, it is likely to be at the level of translation. (**C**) The intrinsic RNA binding capacity of NHL-2 could enable it to regulate a large number of transcripts in the soma and germline, both positively and negatively. This activity could be influenced by additional binding partners, and we speculate that it may occur in P bodies (soma) and P granules (germline) based on the localization pattern of NHL-2.

DOI: https://doi.org/10.7554/eLife.35478.022

for one such factor. Third, the role for NHL-2 in biogenesis of a subset of germline 22G-RNAs and the effector steps of somatic miRNAs demonstrate differential roles for this intriguing protein in the germline versus soma, and points to differences in protein binding partners in each of these tissues that should be examined further.

Notably, many of the up-regulated genes in *nhl-2(ok818)* mutants were the targets of 22G-RNAs, yet these genes did not display alterations in the levels of 22G-RNAs. These data point to a role for NHL-2 in regulating transcript stability, in cooperation with the WAGOs, and in opposition to CSR-1, and suggest a combinatorial regulatory mechanism that engages both small RNA pathways and *bona fide* RBPs such as NHL-2. The fact that *nhl-2(ok818)* mutants display temperature dependent fertility defects is consistent with several small RNA pathway factors, including both the piRNA/WAGO and the CSR-1 pathways and points to NHL-2 as a co-factor and/or co-regulator required for optimal pathway activity under stressful conditions. Although NHL-2 was not identified previously as

a factor in the piRNA or 22G-RNA pathways, our results exemplify the power of synthetic genetic screens to identify accessory factors involved in the optimal function these pathways.

In light of our data, we propose that NHL-2 acts as a hub of gene regulation, where it works cooperatively with core factors in a diverse set of pathways that are central to both somatic and germline gene regulation (*Figure 8*). NHL-2 localizes to several ribonucleoprotein structures involved in RNA regulation, including P granules in the germline, CGH-1 granules in the gonad core, and cytoplasmic P bodies in the soma, placing it in key cellular positions to regulate multiple facets of gene expression and RNA metabolism throughout development. For instance, P granules and related Mutator Foci (to which NHL-2 does not appear to localize) are thought to be important sites for the synthesis of 22G-RNAs and 22G-RNA-mediated gene regulation, as RdRP components and Argonautes, including CSR-1, WAGO-1, and HRDE-1 also localize to these sites (*Phillips et al., 2012*; *Claycomb et al., 2009*). CGH-1 granules of the gonad core have a complex interplay with P granules and are enriched for translational regulators, including CGH-1 and IFET-1. The localization of NHL-2 to both P and CGH-1 granules in the germline supports the observation that NHL-2 and CGH-1 regulate the stability of a shared set of germline/oogenesis transcripts, and opens the possibility that NHL-2 and CGH-1 could regulate the translation of a shared set of targets. The intrinsic RNA binding capacity of NHL-2 via its NHL domain also makes it plausible that NHL-2 could regulate its own set of target mRNAs, independent of small RNA pathways or CGH-1. In fact, we observed that a large number of non-gonadal transcripts are regulated, both positively and negatively, in an NHL-2-dependent manner, and these transcripts are a separate group from those targeted by 22G-RNAs or CGH-1. Ultimately, quantitative proteomic studies, ribosome profiling in *nhl-2(ok818)* mutants, and NHL-2 RNA-IP or CLIP-seq experiments will enable us to identify the full repertoire of mechanisms of NHL-2 regulation.

## Conclusions

In conclusion, we characterized the roles of NHL-2 in the germline and showed it localizes to P granules and impacts a subset of 22G-RNAs in both the CSR-1 and WAGO/HRDE pathways. This germline role in small RNA biogenesis is distinct from its role in the miRNA pathway in the soma, and implicates NHL-2 in RdRP activity in the germline. Interestingly, also NHL-2 was required for the nuclear RNAi pathway, suggesting that NHL-2 is a promiscuous co-factor of multiple distinct, but related small RNA pathways. NHL-2 displays intrinsic RNA binding ability via its NHL domain and thus is capable of binding and regulating the stability or translation of a large number of germline transcripts in a small RNA-independent manner. NHL-2 may exemplify a new class of co-factor that is required for optimal activity of small RNA pathways (both miRNA and 22G-RNA pathways). Although this type of co-factor appears to be extremely important for the fidelity and robustness of developmental gene expression programs, additional examples of such co-factors are not likely to be identified by forward genetic screens, but may be revealed through similar screens in sensitized backgrounds. Overall, our study highlights the complex roles of the TRIM-NHL protein NHL-2 and lays the foundation for deeper mechanistic insights into how these versatile and conserved proteins act to regulate gene expression in various tissues.

## Materials and methods

### Worm strains

Strains used in this study were Bristol N2 as the wild-type, PRB310 (*nhl-2(ok818) III*), YY186 (*nrde-2(gg091) II*), WM49 (*rde-4(ne301) III*), WM182 *csr-1(tm892)* IV/nT1 [*unc-?(n754) let-?*] (IV;V), WM206 (*drh-3(ne4253)* I), PRB316 (*nhl-2(ok818)*III; *mjIs31[ppie-1::gfp::h2b]*II, SX1442 (*mjIs31[ppie-1::gfp::h2b]*II; *nrde-2(mj168)*II), SX2127 (*mjIs31[ppie-1::gfp::h2b]II; hrde-1(tm1200)*III) and VT3653 (*nhl-2(ma372 ma399) III*). The *nhl-2(ma372 ma399)* null strain was generated by deleting the entire NHL-2 coding sequence and all selectable markers from a GFP-tagged *nhl-2* locus (*nhl-2(ma372)*) (*McJunkin and Ambros, 2017*) to yield *Pnhl-2::GFP::nhl-2 3'UTR*. Guide RNAs were ordered as AltR guides from IDT and injected as preassembled RNP with Cas9 protein (*Paix et al., 2014*). Protospacer sequences were CGGAATGGACGAGCTCTACA (end of GFP CDS) and ATCAGCAAG TTTCAGTGAGG (end of NHL-2 CDS). The resulting deletion allele creates the following junction where the underlined region is a spurious insertion between GFP and the *nhl-2* 3'UTR:

acccacggaatggacgagcttacccaaaggaggttaccccaattcct. Some strains were obtained from the *C. elegans* Genetics Centre (CGC, USA) and cultured under standard conditions (*Brenner, 1974*).

## RNAi

RNAi was performed using the feeding method (*Timmons and Fire, 1998*). Each bacterial feeding clone was grown overnight in 2xTY media with 100 μg/ml ampicillin and seeded onto NGM plates with 100 μg/ml ampicillin and 4 mM IPTG. Approximately 10 synchronized L1 animals of each strain were then pipetted onto each plate.

## Transgenic worms

To make transgenic *nhl-2(ok818)* worms expressing a rescuing GFP::NHL-2 fusion protein (*Hammell et al., 2009*), young adult worms were in injected with the plasmids pCMH720 and pRF4 (the *rol-6* marker) at 30 ng/μl and 100 ng/μl respectively. Three independent extrachromosomal lines were analyzed.

## Brood size assay, Transgenerational and Mortal germline assays

Brood size assays were performed on animals fed *E. coli* OP50 or bacteria expressing RNAi clones. Synchronized populations of each strain were grown at 20°C and 25°C until the fourth larval stage (L4) and then individual L4 animals were then placed onto pre-seeded NGM plates and returned to their respective temperatures. Animals were then transferred to new plates every 12 hr and plates were scored for progeny after 48 hr. This process was repeated until animals failed to lay new progeny. Total progeny included viable progeny and unhatched embryos, while embryonic lethality was scored separately as unhatched embryos. For transgenerational brood size analysis and Mrt assays, wild-type and homozygote *nhl-2(ok818)* worms were grown at 20°C for at least five generations and then shifted to 25°C as L1. In the transgenerational assay, F1 and subsequent generations were grown at either 25°C or 20°C for successive generations and total brood size and embryonic lethality scored. For the Mrt assay animals were maintained at 25°C and six L2/3 animals were transferred at each generation to individual plates and counting all progeny.

## Nuclear RNAi assay

The *lir-1* RNAi clone was obtained from the ORFome RNAi library and sequence to confirm the identity. Synchronised L1 worms were plated on *lir-1* or control RNAi plates and grown at 25°C for 72 hr and then inspected for arrested or dead worms.

## RNAi inheritance assays

Synchronised L1 *hl-2(ok818)* animals were plated onto *gfp* or control (*Arabidopsis thaliana* gene *Lhcb4.3* (pCB19), which has no homology to any *C. elegans* gene) RNAi bacteria. Animals were scored 3 days later for the presence of GFP (oocytes were visualised by fluorescence microscopy using a 63x objective), with silenced animals bleached treated to collect progeny which were transferred OP50 plates for subsequent generations.

## Gonad dissection and immunostaining

One-day-old adults were anesthetized (0.01% tetramisole) and gonads dissected, snap frozen in liquid nitrogen and processed as described in *Navarro et al., 2001*. Rabbit anti-NHL-2 antibodies was raised to the peptide RHESPATSTNNTQNS (GL Biochem, China). Anti-NHL-2 and anti-CGH-1 primary antibodies were incubated overnight at 4°C and secondary antibodies for 2 hr at 20°C. Slides were washed twice in PBS/0.1% Tween-20 for 10 min at 20°C and mounted for imaging using Dako Fluorescent Mounting Media (Dako, Denmark). Fluorescent images were acquired using an inverted Olympus IX81 × 2 UCB microscope (Olympus, Tokyo, Japan) attached to an X-cite series 120Q fluorescent light box.

## Immunoprecipitation and GST Pull-down assay

Full-length NHL-2 coding sequence was cloned into modified pDEST-Magic vector pTH6838, resulting in an N-terminally-tagged NHL-2 expression construct. *Escherichia coli* C41 cells (Lucigen) were transformed using NHL-2 expression construct or empty plasmid for tag-only control experiments.

Protein expression was induced by adding IPTG (1 mM final) to overnight log phase cell culture at 37°C, and continued for 3.5–4 hr. Cell lysates were prepared by sonication, and the added to GST resin for binding. After washing off non-specific binders, tagged NHL-2 or GST protein was eluted off the resin using 250 mM NaCl, 50 mM Tris-HCl (pH 8.8), 30 mM reduced glutathione, 10 mM BME and 20% Glycerol. Protein concentration and purity were estimated by SDS-PAGE using standard procedure. Details for NHL-2 expression and purification processes can be found elsewhere (*Ray et al., 2017*; *Ray et al., 2013*).

*C. elegans* lysate was prepared from gravid wild-type adults in 100 mM NaCl, 50 mM Sodium phosphate (pH 7.5), 0.05% triton and a combination of protease (Roche) and phosphatase inhibitors. For each binding experiment, 5 mg of worm lysate was pre-cleared using 25 ul of GST resin. 8 ug of GST-tagged NHL-2 or 4 ug of GST protein was added to pre-cleared lysate containing fresh 25 ul of GST resin in 1 ml final reaction volume. The mixture was incubated at 4°C for 3 hr. Proteins unbound to the resin were removed by washing the resin 5 times with 50 ul 100 mM NaCl, 50 mM sodium phosphate, 0.05% triton buffer. Bound proteins were eluted by adding 50 ul of 1X SDS gel loading buffer to the resin followed by heating the samples at 70°C for 1 min. Candidate NHL-2 binding partners were separated on SDS- PAGE and probed by western blots using published *C. elegans* antibodies. Tubulin was used for loading controls between GST-NHL-2 and tag only experiments.

## Protein expression, RNAcompete and Fluorescence Anisotropy

RNA pool generation, RNAcompete pulldown assays and microarray hybridizations were performed as previously described (*Ray et al., 2017*; *Ray et al., 2013*; *Ray et al., 2009*). Briefly, NHL domain of NHL-2 (residues 625–1032) was expressed as a GST-tagged fusion protein using the pGEX vector (pTH6838) (GE Healthcare). GST-NHL-2 (20 pmoles) and RNA pool (1.5 nmoles) were incubated in 1 mL of Binding Buffer (20 mM HEPES pH 7.8, 80 mM KCl, 20 mM NaCl, 10% glycerol, 2 mM DTT, 0.1 μg/μL BSA) containing 20 μL glutathione sepharose 4B (GE Healthcare) beads (pre-washed 3 times in Binding Buffer) for 30 min at 4°C, and subsequently washed four times for two minutes with Binding Buffer at 4°C. One-sided Z-scores were calculated for the motifs as described previously (*Ray et al., 2013*).

For fluorescence anisotropy assays, *Escherichia coli* BL21(DE3) expresing GST-NHL-2 (729–1032) were grown to an $OD_{600}$ = 0.6–0.8 at 37°C and induced with the addition of 0.5 mM IPTG for 16 hr at 23°C. Cells were resuspended in lysis buffer (100 mM Tris.Cl pH 7.0, 5 mM EDTA, 5 mM DTT supplemented with 1 x 'cOmplete' Protease Inhibitor Cocktail (Roche), passed through a French press 4 times at 16,000–18,000 psi. and the lysate clarified by centrifugation at 8,000 *g* for 30 min. The GST-NHL-2 fusion protein was purified using a 5 mL GSTrap FF based on the manufactures specification (GE healthcare). Alterations to the protocol included washing the column 5 x volumes of binding buffer (1 x PBS pH 7.3) containing an additional 2 M NaCl to remove any DNA and RNA bound to NHL-2, followed by five column volumes of binding buffer. To elute the bound protein, the column was washed with elution buffer (1x PBS, 10 mM reduced glutathione pH 8.0) and eluted fractions run on a SDS-PAGE gel to determine which fractions GST-NHL-2 eluted in, these fractions were then pooled and concentrated to be then further purified using size exclusion on the HiLoad 16/60 Superdex 200. The NHL domain purity was >95% by SDS-PAGE and was quantitated by $OD_{280}$ using an extinction coefficient of 61935 $M^{-1}cm^{-1}$ (*Pace et al., 1995*).

To examine the RNA-binding ability of the NHL domain, a 12-point serial dilution (0–52.81 μM) of GST and GST-NHL-2 NHL domain was incubated with 1 nM 5'-Fluorescein labelled 17 mer poly-U single-stranded RNA (Dharmacon GE, USA) in assay buffer (50 mM NaCl, 20 mM $NaPO_4$, 2 mM $MgCl_2$, 1 mM DTT, 10% glycerol pH 7.4) for 15 min at room temperature in 96-well non-binding black plates (Greiner Bio-One). Fluorescence anisotropy was measured in triplicate using PHERAstar FS (BMG) with FP 488-520-520 nm filters. Data were corrected for anisotropy of RNA alone samples, and then fitted to a one-site binding model using the Equation, $A = (A_{max} [L])/(K_D+[L])$, where A is the corrected fluorescence anisotropy; $A_{max}$ is maximum binding fluorescence anisotropy signal, [L] is the NHL concentration, and $K_D$ is the dissociation equilibrium constant. $A_{max}$ and $K_D$ were used as fitting parameters and nonlinear regression was performed using SigmaPlot 13.0.

## Small RNA cloning and data analysis

Total RNA was collected from wild-type and *nhl-2(ok818)* mutants that were grown to gravid adulthood at either 20°C or 25°C on OP50 *E. coli*. We isolated small RNAs and prepared libraries for Illumina sequencing using a previously described cloning strategy (*Tu et al., 2015*; *Gu et al., 2009*). The only minor modification to these protocols was that we used 0.1X the normal amount of Tobacco Acid Pyrophosphatase (enabling us to capture a greater proportion of miRNAs, while still sufficiently recovering 22G-RNAs). Small RNA analysis was conducted using custom Shell and Perl (5.10.0) scripts (*Supplementary file 4*) (*Tu et al., 2015*). The reference genome of *C. elegans* and genomic annotations were downloaded from WormBase (Release WS230) (*Yook et al., 2012*). The snoRNAs annotated in GenBank were also included in the category of non-coding RNAs (ncRNAs). The sequences of pre-miRNAs and mature miRNAs were fetched from miRBase (Release 19). After removing the barcodes from small RNA-seq reads for each sample, the insert size were extracted by allowing at most one mismatch in the first 6 nt of the 3′ linker (CTGTAG). We first excluded the reads that could be aligned to the ncRNAs, and then aligned the remaining reads to the genome using Bowtie (*Langmead et al., 2009*) without mismatches. Small RNA abundance were normalized to the sequencing depth (the sum of genome-mapping and junction-mapping reads but those known ncRNAs) as reads per million (rpm). We used the same lists of CSR-1 targets, WAGO-1 targets, etc. as in (*Tu et al., 2015*), and the same criteria to define the genes depleted of 22G-RNAs in *nhl-2 (ok818)* mutants, that is at least 2-fold depletion in mutants than wild-type and at least 10 rpm reads in wild-type. The significance of the overlap between gene sets was calculated by hypergeometric test. The miRNA targets were predicted using TargetScan (*Lewis et al., 2005*).

## mRNA-Seq

Total RNA was extracted from triplicate samples of wild-type and *nhl-2(ok818)* mutants that were grown to gravid adulthood at either 20°C or 25°C on OP50 *E. coli*. mRNA-seq libraries were generated and sequenced at the Donnelly Sequencing Centre (University of Toronto) using the Illumina Tru-Seq Stranded mRNA Library Construction Kit and the HiSeq2500 sequencer. The mRNA-seq reads were mapped to the reference genome by Tophat (*Trapnell et al., 2009*) with default parameters. For each gene, we counted the number of the 22G-RNA reads that are antisense to the corresponding gene transcript. With the counts, edgeR was used to integrate the three replicates of *nhl-2* mutants versus the three replicates of wild-type. Then, a gene is called to be depleted of 22G-RNAs in *nhl-2* mutants if the corresponding FDR and fold-change (FC) by edgeR satisfy FDR < 0.05 and FC < ½, and the gene has at least 10 reads per million (rpm) antisense 22G-RNAs reads in at least one of three replicates of wild-type.

## Accession numbers

All small RNA and mRNA Illumina sequencing data have been submitted to the NCBI's Sequence Read Archive (SRA), and are included under project accession number SRP115391.

## Acknowledgements

Strains were provided by the *Caenorhabditis Genetics* Center, which is funded by the National Institutes of Health Office of Research Infrastructure Programs (P40 OD010440). We thank Inga Loedige for generously providing the GST-NHL-2 NHL domain construct used for fluorescence anisotropy. We thank Christopher Wedeles for critical reading of the manuscript. G.M.D was supported by a Monash University PhD scholarship. KM was funded by NIDDK Intramural Research Program (DK075147), QM and TRH were funded by CIHR (grant MOP-125894), JMC was funded by the Canada Research Chairs program, the CIHR (grants MOP-274660 and CAP- 262134), and the University of Toronto Department of Molecular Genetics and Connaught Fund. S T and ZW were partly funded by the NIH grant HD078253. PRB was funded by the National Health and Medical Research Council of Australia (NHMRC) (grant number 0606575).

## Additional information

### Funding

| Funder | Grant reference number | Author |
|--------|------------------------|--------|
| National Institute of Diabetes and Digestive and Kidney Diseases | DK075147 | Katherine McJunkin |
| Canadian Institutes of Health Research | MOP-125894 | Quaid D Morris |
| Canadian Institutes of Health Research | MOP-274660 | Julie M Claycomb |
| Connaught Fund | | Julie M Claycomb |
| Canadian Institutes of Health Research | CAP- 783 262134 | Julie M Claycomb |
| Canada Research Chairs | | Julie M Claycomb |
| University of Toronto | | Julie M Claycomb |
| National Institutes of Health | HD078253 | Zhiping Weng |
| National Health and Medical Research Council | 0606575 | Peter R Boag |

The funders had no role in study design, data collection and interpretation, or the decision to submit the work for publication.

### Author contributions

Gregory M Davis, Conceptualization, Data curation, Formal analysis, Investigation, Methodology, Writing—original draft, Writing—review and editing; Shikui Tu, Data curation, Software, Formal analysis, Methodology; Joshua WT Anderson, Data curation, Formal analysis, Investigation, Methodology, Writing—review and editing; Rhys N Colson, Data curation, Formal analysis, Methodology, Writing—review and editing; Menachem J Gunzburg, Data curation, Formal analysis; Michelle A Francisco, Debashish Ray, Monica Z Wu, Formal analysis, Methodology; Sean P Shrubsole, Data curation, Formal analysis, Methodology; Julia A Sobotka, Uri Seroussi, Robert X Lao, Formal analysis; Tuhin Maity, Data curation, Methodology; Katherine McJunkin, Resources; Quaid D Morris, Timothy R Hughes, Supervision, Writing—review and editing; Jacqueline A Wilce, Conceptualization, Data curation, Formal analysis, Supervision, Methodology, Writing—original draft, Writing—review and editing; Julie M Claycomb, Conceptualization, Resources, Formal analysis, Supervision, Funding acquisition, Writing—original draft, Project administration, Writing—review and editing; Zhiping Weng, Conceptualization, Formal analysis, Supervision, Funding acquisition, Methodology, Writing—review and editing; Peter R Boag, Conceptualization, Resources, Data curation, Formal analysis, Supervision, Funding acquisition, Writing—original draft, Project administration, Writing—review and editing

### Author ORCIDs

Peter R Boag (ORCID) https://orcid.org/0000-0002-0889-0859

### Decision letter and Author response

Decision letter https://doi.org/10.7554/eLife.35478.031
Author response https://doi.org/10.7554/eLife.35478.032

## Additional files

### Supplementary files

• Supplementary file 1. Small RNA sequencing statistics of four samples in *nhl-2(ok818)* mutants and wild-type animals at 25°C or 20°C, Related to *Figures 4* and *6*.
DOI: https://doi.org/10.7554/eLife.35478.023

• Supplementary file 2. Genes altered in small RNA and mRNA sequencing datasets generated and investigated in this paper, Related to *Figures 4* and *6*.
DOI: https://doi.org/10.7554/eLife.35478.024

• Supplementary file 3. Master file of small RNA and mRNA-seq data used in this study.
DOI: https://doi.org/10.7554/eLife.35478.025

• Supplementary file 4. Source code file for the custom Shell and Perl (5.10.0) scripts used for Small RNA analysis.
DOI: https://doi.org/10.7554/eLife.35478.026

• Transparent reporting form
DOI: https://doi.org/10.7554/eLife.35478.027

### Data availability

All small RNA and mRNA Illumina sequencing data have been submitted to the NCBI's Sequence Read Archive (SRA), and are included under project accession number SRP115391.

The following dataset was generated:

| Author(s) | Year | Dataset title | Dataset URL | Database and Identifier |
|---|---|---|---|---|
| Shikui Tu | 2017 | Caenorhabditis elegans Raw sequences reads NHL-2, Aug 12'17 | https://www.ncbi.nlm.nih.gov/sra/?term=SRP115391 | NCBI Sequence Read Archive, SRP115391 |

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
