## [Decision Letter]

Thank you for submitting your article "The TRIM-NHL protein NHL-2 is a Novel Co-Factor of the CSR-1 and HRDE-1 22G-RNA Pathways" for consideration by *eLife*. Your article has been reviewed by Jessica Tyler as the Senior Editor, a Reviewing Editor (Oliver Hobert), and two reviewers. The following individual involved in review of your submission has agreed to reveal his identity: John K Kim (Reviewer #2).

The reviewers have discussed the reviews with one another and the Reviewing Editor has drafted this decision to help you prepare a revised submission. In brief, the depth and breadth of your analysis of the very interesting nhl-2 gene was much appreciated. However, there were a number of concerns that will need to be addressed. One major issue is the overreliance on a single mutant allele (Reviewer #1), the other is the need to tie together some of the observations into a more coherent story (first few comments of Reviewer #2). It is important to not just address these major points editorially, but to conduct the requested experiments, which we do not believe to be too time-consuming. Most of the other points are editorial in nature.

*Reviewer #1:*

In this manuscript by Davies et al., the authors characterize the function on the TRIM-NHL factor NHL-2 in reproduction, small RNA biology, and gene regulation in the germline. The study provides evidence linking NHL-2 to germline chromatin modification, chromosome organization, germ cell function, embryonic viability, nuclear RNAi, and various endogenous germline small RNA pathways. A detailed analysis of the effect of temperature on the role of NHL-2 in various small RNA pathways is provided. Finally, the manuscript provides evidence that NHL-2 is likely a polyU RNA binding protein that contributes extensively to gene regulation in the germline although, surprisingly, this gene regulation appears unrelated to the role of NHL-2 in small RNA pathways. The paper is well-written, and the data are clearly displayed. The subject matter is of broad interest to *eLife* readership. The paper, however, mostly fails to explain how the various functions of NHL-2 are integrated or define the mechanism by which NHL-2 mediates its many biological functions. Technical concerns also mitigate enthusiasm (see below). For these reasons, I do not believe the manuscript is appropriate for *eLife* at this time.

Technical concerns. The manuscript relies entirely upon the analysis of a single nhl-2 deletion strain. Additional mutant alleles of nhl-2 will need to be tested for key phenotypes before any of the conclusions reached in the paper are justified. Alternatively, rescue or nhl-2 RNAi knockdown could be included. Minor. nhl-2 pathway studies employ candidate gene RNAi knockdown in wild-type or nhl-2 mutant strains to test for genetic interactions. Elsewhere in the paper the authors show that NHL-2 contribute to RNAi-mediated gene silencing, which will complicate interpretation of pathway data. Classical genetic studies will be needed to assess potential genetic interactions.

*Reviewer #2:*

Summary:

This paper investigates the role of the TRIM-NHL protein NHL-2 as a cofactor in the 22G-RNA pathways in *C. elegans*. NHL-2 is involved in the regulation of lsy-6 and let-7 associated miRISC activity (Hammell et al., 2009) and in the sex determination pathway (McJunkin and Ambros, 2017). The function of TRIM-NHL proteins was hypothesized to be driven by protein-protein interaction, like in the miRNA pathway. However, TRIM-NHL proteins can also directly bind RNA (Laver et al., 2015). This paper provides insights into the mechanistic details of NHL-2 function mainly in the germline.

NHL-2 colocalized with CGH-1 in both germline P granules and the germline cytoplasmic core. nhl-2(ok818) showed defects in brood size and increased embryonic lethality. nhl-2(ok818) oocytes also exhibited defects in chromosome segregation at diakinesis with greater than 6 DAPI stained bodies and chromosomal aggregations. A genome-wide RNAi screen revealed that knockdown of several 22G-RNA factors like drh-3, csr-1, ekl-1 and cde-1 in nhl-2(ok818) animals resulted in enhanced chromosomal segregation defects. nhl-2(ok818) animals were resistant to lir-1 RNAi mediated larval arrest and displayed a transgenerational mortal germline phenotype, indicating a possible role for NHL-2 in germline and somatic nuclear RNAi. NHL-2 was found to physically interact with DRH-3, HRDE-1 and CSR-1. Small RNA-seq in nhl-2(ok818) animals showed that there was a decrease in 22G-RNA and 26G-RNA populations. The genes corresponding to these small RNAs showed significant overlap with HRDE-1, CSR-1 and WAGO-1 targets. Analysis of the distribution of the 22G-RNAs indicated that the depletion was observed mainly in the 5' end of the CSR-1 target genes, suggesting that NHL-2 influences the processivity of the RdRP complex. in vitro binding assays showed that NHL-2 is capable of binding U-rich RNA. mRNA-seq in nhl-2(ok818) animals showed that a several genes had misregulated transcript levels. However, a very small proportion of genes that show alteration in 22G levels exhibit upregulated transcript levels. Significant overlap was found between the upregulated genes in nhl-2(ok818) and WAGO-1, HRDE-1 and CSR-1 targets indicating that NHL-2 regulates transcript levels in a small RNA independent manner, probably by binding mRNAs directly.

This is the first paper to elucidate the detailed role of NHL-2 in the germline and thus makes an important contribution to the field. The authors use both genetic and biochemical techniques to show convincingly that NHL-2 may act as a cofactor in the 22G-RNA pathway, not only by influencing the processivity of RdRPs but also by regulating stability of transcripts.

Essential revisions:

1) Figure 2 has elegantly designed experiments. However, as aberrant chromosomal aggregation is one of the outcomes of 22G misregulation, enhancement of this phenotype with knockdown of 22G factors in nhl-2(ok818) animals, suggests that nhl-2 is probably involved in a parallel pathway to the 22G-RNA pathway (and not in the same pathway, as mentioned in the paper) that together result in proper chromosomal segregation. Similarly, looking at the H3K9me2 distribution in nhl-2(ok818) with drh-3 or csr-1 RNAi will provide more insight. This also helps bring the paper together, when data from Figure 7 is considered.

2) Given that HRDE-1 is involved in transgenerational germline silencing, it will be interesting to subject nhl-2(ok818) animals to transgenerational reporter silencing assays (for instance with a Ppie1::GFP reporter system).

3) It may be useful to include a scatter plot in Figure 5 to clearly indicate the population of 22G and 26G-RNAs that get depleted in nhl-2(ok818).

4) In Figure 5D, for WAGO-1 targets, a depletion at the 5' end is also seen, but not mentioned in the text anywhere. Is this not significant?

5) It will be interesting to see if the genes with altered transcript levels in nhl-2(ok818) overlap with CSR-1 mediated slicer target genes or CSR-1 mediated transcriptional activation (Cecere et al., 2014).

6) For Figure 5 and Figure 7, make sure that the references for all the data sets used are mentioned. For instance, the reference for the data set associated with the RdRP complex mutants has not been mentioned.

7) In Figure 1—figure supplement 1, schematic for NCL-1 is not shown, but mentioned in the legend. It may also be useful to include a legend relating the color and the corresponding domain.

[Editors' note: further revisions were requested prior to acceptance, as described below.]

Thank you for submitting your article "The TRIM-NHL protein NHL-2 is a Co-Factor in the Nuclear and Somatic RNAi Pathways in *C. elegans*" for consideration by *eLife*. Your article has been reviewed by Jessica Tyler as the Senior Editor, a Reviewing Editor, and two reviewers. The following individual involved in review of your submission has agreed to reveal his identity: John K Kim (Reviewer #2).

The reviewers have discussed the reviews with one another and the Reviewing Editor has drafted this decision to help you prepare a revised submission.

While the reviewers appreciate the efforts that have been undertaken to improve the manuscript, they felt that a key critical concern has not been sufficiently addressed, namely the validation that the multitude of phenotypes ascribed to the loss of nhl-2 are indeed due to loss of the gene. Rescue has only been provided for one specific (quite uninformative) phenotype, L1 arrest. And the RNAi phenocopy analysis is also limited. The reviewers and editor agree that it is essential that the authors analyze a second allele of nhl-2 and recommend the usage of a null allele described by the Ambros lab (https://www.ncbi.nlm.nih.gov/pubmed/28279983). This analysis should not be limited to a single phenotype, but to all of the relevant phenotypes that the authors describe for the currently existing allele.

---

## [Author Response]

The reviewers have discussed the reviews with one another and the Reviewing Editor has drafted this decision to help you prepare a revised submission. In brief, the depth and breadth of your analysis of the very interesting nhl-2 gene was much appreciated. However, there were a number of concerns that will need to be addressed. One major issue is the overreliance on a single mutant allele (Reviewer #1), the other is the need to tie together some of the observations into a more coherent story (first few comments of Reviewer #2). It is important to not just address these major points editorially, but to conduct the requested experiments, which we do not believe to be too time-consuming. Most of the other points are editorial in nature.Reviewer #1:In this manuscript by Davies et al., the authors characterize the function on the TRIM-NHL factor NHL-2 in reproduction, small RNA biology, and gene regulation in the germline. The study provides evidence linking NHL-2 to germline chromatin modification, chromosome organization, germ cell function, embryonic viability, nuclear RNAi, and various endogenous germline small RNA pathways. A detailed analysis of the effect of temperature on the role of NHL-2 in various small RNA pathways is provided. Finally, the manuscript provides evidence that NHL-2 is likely a polyU RNA binding protein that contributes extensively to gene regulation in the germline although, surprisingly, this gene regulation appears unrelated to the role of NHL-2 in small RNA pathways. The paper is well-written, and the data are clearly displayed. The subject matter is of broad interest to eLife readership. The paper, however, mostly fails to explain how the various functions of NHL-2 are integrated or define the mechanism by which NHL-2 mediates its many biological functions. Technical concerns also mitigate enthusiasm (see below). For these reasons, I do not believe the manuscript is appropriate for eLife at this time.

We thank the reviewer for their thoughtful read of the manuscript and constructive critiques. We agree that it is somewhat unsatisfying to not completely understand HOW NHL-2 does what it does. This is why we continue to work on understanding the mechanisms by which NHL-2 performs the many functions we have implicated it in via this manuscript. In fact, we currently have a number of structure-function experiments underway to disentangle which portions of NHL-2 are required for which functions. However, we have had these current data for some time and we think it is important to publish the current findings so that others may begin to build upon our work.

Technical concerns. The manuscript relies entirely upon the analysis of a single nhl-2 deletion strain. Additional mutant alleles of nhl-2 will need to be tested for key phenotypes before any of the conclusions reached in the paper are justified. Alternatively, rescue or nhl-2 RNAi knockdown could be included. Minor. nhl-2 pathway studies employ candidate gene RNAi knockdown in wild-type or nhl-2 mutant strains to test for genetic interactions. Elsewhere in the paper the authors show that NHL-2 contribute to RNAi-mediated gene silencing, which will complicate interpretation of pathway data. Classical genetic studies will be needed to assess potential genetic interactions.

We understand and agree with the reviewer’s concern. To address this issue and strengthen our findings, we provide several additional lines of data. First, we generated a strain in which nhl-2 is expressed from the nhl-2 promoter via a multicopy extrachromosomal array. Such extrachromosomal array transgenes are normally silenced in the germline but expressed in the soma. Because of this feature, we were able to test for the ability of this transgene to rescue the nuclear RNAi deficiency of the nhl-2(ok818) mutant (a somatic assay). We show in our new Figure 3B that this transgene rescues the mutant nuclear RNAi deficiency, thus increasing our confidence in attributing this phenotype to loss of nhl-2 in the nhl-2(ok818) strain. Although we are not able to assay the ability of this transgene to rescue germline phenotypes, we know from our own previous RNAi experiments (that we unfortunately did not quantify and thus have not presented herein), and RNAi experiments performed by others (Wormbase) that loss of nhl-2 by RNAi leads to sterility in adult hermaphrodites, consistent with what we have reported here for the mutant. Finally, we examined whether RNAi of nhl-2 leads to a mortal germline phenotype and found that the results recapitulate the Mrt phenotype of the mutant. Collectively, these data provide solid evidence supporting the phenotypes we have observed for the nhl-2(ok818) mutant. [We also note that we attempted to make deletion mutants of nhl-2 via CRISPR-Cas9 editing and recovered alleles that were not useful for our analysis because they produced an in frame truncated NHL-2 protein. Due to time limitations, we were unable to recover a second allele in time.]

Reviewer #2:Summary:This paper investigates the role of the TRIM-NHL protein NHL-2 as a cofactor in the 22G-RNA pathways in C. elegans. NHL-2 is involved in the regulation of lsy-6 and let-7 associated miRISC activity (Hammell et al., 2009) and in the sex determination pathway (McJunkin and Ambros, 2017). The function of TRIM-NHL proteins was hypothesized to be driven by protein-protein interaction, like in the miRNA pathway. However, TRIM-NHL proteins can also directly bind RNA (Laver et al., 2015). This paper provides insights into the mechanistic details of NHL-2 function mainly in the germline.NHL-2 colocalized with CGH-1 in both germline P granules and the germline cytoplasmic core. nhl-2(ok818) showed defects in brood size and increased embryonic lethality. nhl-2(ok818) oocytes also exhibited defects in chromosome segregation at diakinesis with greater than 6 DAPI stained bodies and chromosomal aggregations. A genome-wide RNAi screen revealed that knockdown of several 22G-RNA factors like drh-3, csr-1, ekl-1 and cde-1 in nhl-2(ok818) animals resulted in enhanced chromosomal segregation defects. nhl-2(ok818) animals were resistant to lir-1 RNAi mediated larval arrest and displayed a transgenerational mortal germline phenotype, indicating a possible role for NHL-2 in germline and somatic nuclear RNAi. NHL-2 was found to physically interact with DRH-3, HRDE-1 and CSR-1. Small RNA-seq in nhl-2(ok818) animals showed that there was a decrease in 22G-RNA and 26G-RNA populations. The genes corresponding to these small RNAs showed significant overlap with HRDE-1, CSR-1 and WAGO-1 targets. Analysis of the distribution of the 22G-RNAs indicated that the depletion was observed mainly in the 5' end of the CSR-1 target genes, suggesting that NHL-2 influences the processivity of the RdRP complex. in vitro binding assays showed that NHL-2 is capable of binding U-rich RNA. mRNA-seq in nhl-2(ok818) animals showed that a several genes had misregulated transcript levels. However, a very small proportion of genes that show alteration in 22G levels exhibit upregulated transcript levels. Significant overlap was found between the upregulated genes in nhl-2(ok818) and WAGO-1, HRDE-1 and CSR-1 targets indicating that NHL-2 regulates transcript levels in a small RNA independent manner, probably by binding mRNAs directly.This is the first paper to elucidate the detailed role of NHL-2 in the germline and thus makes an important contribution to the field. The authors use both genetic and biochemical techniques to show convincingly that NHL-2 may act as a cofactor in the 22G-RNA pathway, not only by influencing the processivity of RdRPs but also by regulating stability of transcripts.

We thank the reviewer for this comprehensive summary and constructive comments to improve this manuscript. Your comments were valuable in improving the quality of the science and strengthening our conclusions.

Essential revisions:1) Figure 2 has elegantly designed experiments. However, as aberrant chromosomal aggregation is one of the outcomes of 22G misregulation, enhancement of this phenotype with knockdown of 22G factors in nhl-2(ok818) animals, suggests that nhl-2 is probably involved in a parallel pathway to the 22G-RNA pathway (and not in the same pathway, as mentioned in the paper) that together result in proper chromosomal segregation.

Interpretation of these genetic data is complicated, we agree, thus we have softened statements about the genetic interactions in the manuscript overall. It is correct that there could be several possible interpretations for the data at hand. We see a synergistic phenotype in double mutants/RNAi relative to either single mutant/RNAi (enhanced aggregation) for the diakinesis chromosomal defects (but not for the H3K9me2 spreading, see below). This could indicate that the proteins function in two different pathways and converge on a single molecular/cellular process (chromosome condensation/packaging), as the reviewer suggests. A distinct possibility is that the two proteins are part of a protein complex (acting in the same pathway) that has disrupted function when either is mutated and is even more greatly impaired when both are lost. Because we see a physical interaction between NHL-2 and CSR-1 or DRH-3, this suggests that the synergy could come from disruption of a protein complex within a single pathway. Because we see so many changes in mRNA expression that are independent of small RNA changes in *nhl-2* mutants, it could well be that there is a role for NHL-2 both within the small RNA pathways and within parallel mRNA regulatory pathways. Separation of function alleles and tissue-specific studies will reveal these relationships going forward.

Similarly, looking at the H3K9me2 distribution in nhl-2(ok818) with drh-3 or csr-1 RNAi will provide more insight. This also helps bring the paper together, when data from Figure 7 is considered.

We have included the H3K9me2 data as per the reviewer’s suggestion, and find that for this phenotype, the H3K9me2 spreading is comparable between *nhl-2, csr-1,* and *drh-3* single mutants, and double mutants of *nhl-2* with *csr-1* or *drh-3* show no significant enhancement of the phenotype (Figure 2F, G).

2) Given that HRDE-1 is involved in transgenerational germline silencing, it will be interesting to subject nhl-2(ok818) animals to transgenerational reporter silencing assays (for instance with a Ppie1::GFP reporter system).

Thank you for the suggestion. We generated *nhl-2(ok818)* H2B::GFP reporter worms and examined the inheritance of silencing. We had to conduct these experiments at 20^o^C as the proximal gonad was too disorganized to score at 25°C. *nhl-2(ok818)* H2B::GFP worms showed a modest increase in GFP expression (14%) compared to the *hdre-1(tm1200)* and *ndre-2(mj168*) controls (88% and 84%) (Figure 3F). The comparatively low level of *nhl-2(ok818)* de-silencing may reflect the temperature dependent nature of the *nhl-2(ok818)* phenotypes.

3) It may be useful to include a scatter plot in Figure 5 to clearly indicate the population of 22G and 26G-RNAs that get depleted in nhl-2(ok818).

This seemed like a good suggestion, but it wasn’t as informative as we’d hoped in the end. The best information on populations of 22G-RNAs altered in the mutant comes from Supplementary file 2 and Supplementary file 3, in which we report on the levels of small RNAs for each gene in each data set used. If in the end, you would still like to see these plots, we would be happy to share them with you.

4) In Figure 5D, for WAGO-1 targets, a depletion at the 5' end is also seen, but not mentioned in the text anywhere. Is this not significant?

Good question. This is an interesting situation. It seems that overall, the profile of small RNAs along WAGO targets is noisier than CSR-1 targets (though we are uncertain of what this reflects at this point), and in fact when we performed the same centroid analysis as we did for CSR-1, we found that there is actually a shift in the small RNA pool ever so slightly toward the 5' end of WAGO targets (and this shift is barely statistically significant). We report this observation in the text and in new Figure 5—figure supplement 5 and Figure 5—figure supplement 6. The reason for this shift could be due to technical issues: we captured more of these small RNAs in *nhl-2(ok818)* samples because other small RNAs were missing in these samples (the “filling the cloning space” phenomenon), or it could be biological (perhaps RRF-1 complexes have greater activity on WAGO targets in this scenario because other shared factors like DRH-3, EKL-1 are limiting, for instance).

5) It will be interesting to see if the genes with altered transcript levels in nhl-2(ok818) overlap with CSR-1 mediated slicer target genes or CSR-1 mediated transcriptional activation (Cecere et al., 2014).

Another good question. We should have done this analysis from the start. Now we have included a comparison between in *nhl-2(ok818)* and the CSR-1 GRO-Seq (updated Figure 7C. Upregulated genes in downregulated genes) and the *csr-1* slicer dead (upregulated genes) datasets *nhl-2(ok818)* animals (25°C) showed a significant overlap with both *csr-1* GRO-Seq downregulated and CSR-1 slicer dead upregulated targets. There is minimal overlap between the GRO-Seq downregulated genes and the Slice dead upregulated genes (~15 genes), so it seems that NHL-2 is a consistently negative regulator these different subsets of CSR-1 targets, despite the impact of CSR-1 on these genes (positive vs. negative roles for CSR-1). We would note, however, that these are shockingly small subsets of the CSR-1 targets overall (4932 genes in total), so it is important to keep this in perspective.

6) For Figure 5 and Figure 7, make sure that the references for all the data sets used are mentioned. For instance, the reference for the data set associated with the RdRP complex mutants has not been mentioned.

Thanks for voicing this concern. We could have been more clear about these refs.

throughout the paper, it’s true. We have now added a tab to the Supplementary file 3, reporting all references for the data sets included therein and providing what we hope is a useful legend for each column in this table.

7) In Figure 1—figure supplement 1, schematic for NCL-1 is not shown, but mentioned in the legend. It may also be useful to include a legend relating the color and the corresponding domain.

A schematic for NCL-1 has been included and the proteins domains have been colour coded.

[Editors' note: further revisions were requested prior to acceptance, as described below.]

While the reviewers appreciate the efforts that have been undertaken to improve the manuscript, they felt that a key critical concern has not been sufficiently addressed, namely the validation that the multitude of phenotypes ascribed to the loss of nhl-2 are indeed due to loss of the gene. Rescue has only been provided for one specific (quite uninformative) phenotype, L1 arrest. And the RNAi phenocopy analysis is also limited. The reviewers and editor agree that it is essential that the authors analyze a second allele of nhl-2 and recommend the usage of a null allele described by the Ambros lab (https://www.ncbi.nlm.nih.gov/pubmed/28279983). This analysis should not be limited to a single phenotype, but to all of the relevant phenotypes that the authors describe for the currently existing allele.

We have taken advantage of a newly developed *nhl-2* null allele (*nhl-2(ma372 ma399)*) that removes the entire NHL-2 coding sequence and replaces it with GFP. We have repeated all biological experiments with this new allele and found it strongly phenocopies the *nhl-2(ok818)* allele originally used. We have generated new supplemental figures (list below) that accompany the original figures that used *nhl-2(ok818)*. This new data is referred to it in the appropriate place in the text.

Figure 1—figure supplement 2. *nhl-2(ma372 ma399)* mutants display a temperature sensitive reproduction and embryonic lethality defect.

Figure 2—figure supplement 2. *nhl-2(ma372 ma399)* mutants show defects in diakinetic chromosome organization and H3K9me2 distribution.

Figure 3—figure supplement 1. Analysis of nuclear RNAi pathways in *nhl-2(ma372 ma399)* mutants and *nhl-2(RNAi).*